# Investigating Non-Transitivity in LLM-as-a-Judge

Yi Xu [1]   Laura Ruis [1]   Tim Rocktäschel [1]   Robert Kirk [1][2]

## Abstract

Automatic evaluation methods based on large language models (LLMs) are emerging as the standard tool for assessing the instruction-following abilities of LLM-based agents. The most common method in this paradigm, pairwise comparisons with a baseline model, critically depends on the assumption of transitive preferences. However, the validity of this assumption remains largely unexplored. In this study, we investigate the presence of non-transitivity within the AlpacaEval framework and analyze its effects on model rankings. We find that LLM judges exhibit non-transitive preferences, leading to rankings that are sensitive to the choice of the baseline model. To mitigate this issue, we show that round-robin tournaments combined with Bradley-Terry models of preference can produce more reliable rankings. Notably, our method increases both the Spearman correlation and the Kendall correlation with Chatbot Arena (95.0% → 96.4% and 82.1% → 86.3% respectively). To address the computational cost of round-robin tournaments, we propose **S**wiss-**W**ise **I**terative **M**atchmaking (S**wim**) tournaments, using a dynamic matching strategy to capture the benefits of round-robin tournaments while maintaining computational efficiency.

## 1. Introduction

The growing adoption of large language models (LLMs) as generalist systems for complex, open-ended tasks (OpenAI et al., 2023; Meta AI, 2024b) presents a critical challenge: the lack of a universally accepted gold-standard evaluation. In many cases, multiple valid responses exist for a given task, complicating the establishment of effective benchmarks. Consequently, a new paradigm for evaluating open-ended tasks focuses on quantifying the alignment of LLMs with human preferences (Ouyang et al., 2022) — an

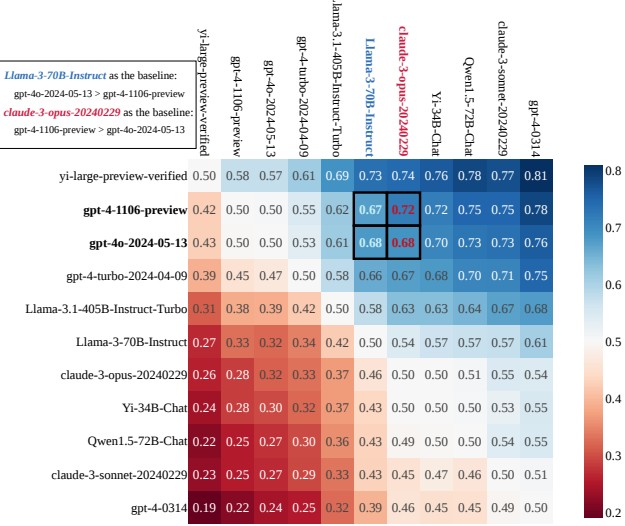

*Figure 1.* **Rankings from baseline-fixed frameworks show high sensitivity to the choice of baseline.** Each entry $(x, y)$ represents the win rate of model $m_x$ against $m_y$, where each column reflects a ranking with the column model as the baseline. Inconsistency emerges when Llama-3-70B and Claude-3-Opus are used as baselines. Appendix C.1 provides the detailed matrix comparing 20 models.

aspect existing automatic metrics cannot adequately assess. However, human evaluation is costly and lacks scalability (Karpinska et al., 2021). As a result, LLM-based evaluators are now widely used to automate the process, with pairwise comparisons proving particularly effective in aligning with human ratings (Liusie et al., 2024; Liu et al., 2024; Chiang et al., 2023; Li et al., 2023; Lin et al., 2025; Zheng et al., 2023; Samvelyan et al., 2024; Khan et al., 2024).

The typical pipeline for LLM-based automatic evaluation frameworks is using pairwise comparisons between a target model and a fixed baseline model, where an oracle model serves as the judge. By calculating the relative win rate against the baseline model, such comparisons enable ranking target models. However, it is unclear whether using a fixed baseline provides consistent results. If the judge exhibits non-transitive preferences, such as favoring A over B, B over C, but C over A, the resulting rankings can become sensitive to the choice of the baseline model (Figure 2).

[1]AI Centre, UCL [2]UK AI Security Institute. Correspondence to: Yi Xu <y.xu.23@ucl.ac.uk>.

*Proceedings of the 42nd International Conference on Machine Learning*, Vancouver, Canada. PMLR 267, 2025. Copyright 2025 by the author(s).

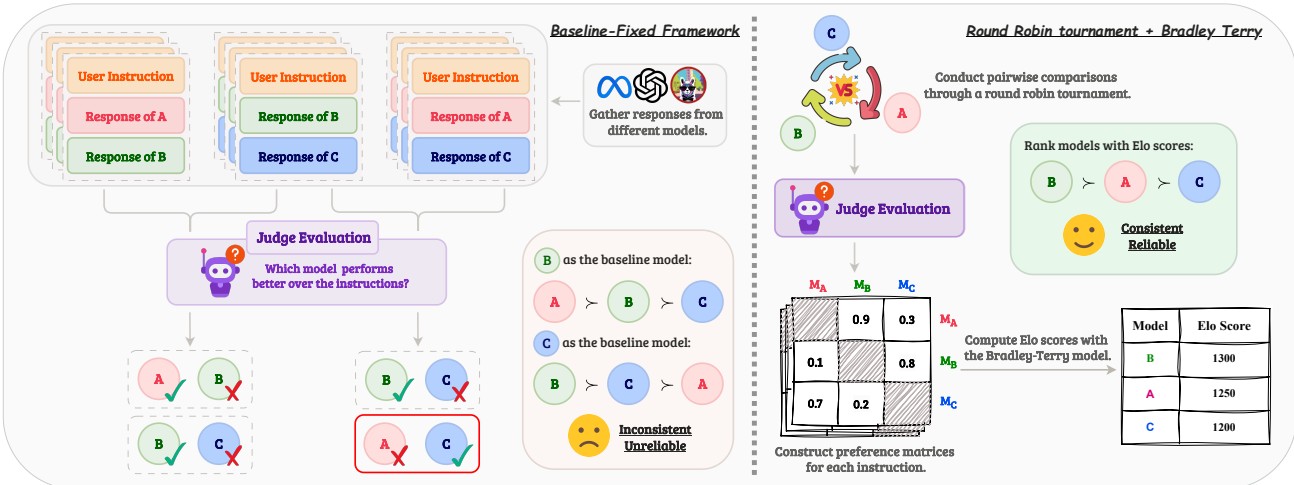

*Figure 2.* (Left) Inconsistent rankings are observed in baseline-fixed frameworks based on pairwise comparisons due to non-transitivity in the judge's evaluations. Different choices of baselines can lead to varying rankings, undermining the reliability and robustness of this approach. (Right) We propose a round-robin tournament framework where all models are compared pairwise. The results are used to capture non-transitivity in the judge's evaluations and score models using the Bradley-Terry model. This method produces rankings that are more robust and better aligned with human evaluation.

In this work, we investigate the existence and impact of non-transitivity within AlpacaEval (Li et al., 2023), which has been largely overlooked in previous work. AlpacaEval is a popular pairwise comparison framework that employs GPT-4-Turbo as the fixed baseline model. We introduce Soft Non-Transitivity Deviation (SNTD) as a metric to measure the degree of soft non-transitivity in the judge's continuous preferences and find that LLMs exhibit both hard and soft non-transitive preferences. Additionally, previous studies have demonstrated that LLMs often exhibit various biases (Gallegos et al., 2024) such as position bias (Zheng et al., 2023; Wang et al., 2024; Zhou et al., 2024b), which can lead to spurious correlations in the judge's preferences. We show that the occurrence of non-transitivity is jointly influenced by position bias and the judge model's inherent non-transitive reasoning abilities.

To address the above, we propose the use of round-robin tournaments in the pairwise comparison setting, overcoming the need for a fixed baseline model. We subsequently apply the Bradley-Terry model (Bradley & Terry, 1952) to score models based on tournament outcomes, yielding a more consistent ranking compared to baseline-fixed ranking. To address the computational cost in the round-robin tournament, we propose Swiss-Wise Iterative Matchmaking (SWIM) tournaments to improve efficiency while preserving the robustness of model comparisons.

Our contributions are as follows: 1) **We show that LLMs exhibit non-transitive preferences when performing pairwise comparisons.** Additionally, we observe that the aggregation of instruction-level non-transitive relationships culminates in model-level non-transitivity (Figure 1). We demonstrate that such non-transitivity makes the ranking highly sensitive to the choice of the baseline model. Changing the baseline model makes the rank order inconsistent and unstable, highlighting the importance of proposing new ranking methods. 2) **We find that while position bias significantly contributes to non-transitivity, it is not the sole cause.** Our experiments confirm that position switching outperforms random assignment in mitigating position bias for stronger judges when using continuous values for judge's preferences, with reductions ranging from 17% to 44%. 3) **We demonstrate that applying round-robin tournaments combined with the Bradley-Terry model reduces the impact of non-transitivity, resulting in more robust rankings.** This method also aligns better with human evaluations of model rankings in Chatbot Arena. Finally, we introduce SWIM, an efficient method for adding models with nearly identical performance compared to naive round-robin tournaments.

## 2. Related Work

**LLM-as-a-Judge.** The LLM-as-a-Judge (Zheng et al., 2023) evaluation method leverages frontier models to rank responses to open-ended queries without explicit ground-truths. A common approach involves using a fixed baseline model for pairwise comparisons to assess the performance of the target model, as seen in frameworks such as VicunaEval (Chiang et al., 2023), AlpacaEval (Li et al., 2023), and Arena-Hard (Li et al., 2024). The target models are then ranked on the basis of their win rates against the baseline.

However, an implicit assumption in these frameworks is that transitivity holds in preference judgments, which has not been empirically verified. Transitivity requires that if an LLM judge prefers model $m_A$ over $m_B$ and $m_B$ over $m_C$, it must consequently prefer $m_A$ over $m_C$. Violations of transitivity can result in unstable rankings that undermine the evaluation framework's reliability (Figure 2). To address this gap, we examine the robustness of current LLM ranking methodologies by extending the AlpacaEval framework to investigate the existence of non-transitivity, aiming to establish a more rigorous foundation for the LLM evaluation system.

**Non-Transitivity in Zero-sum Games.** Prior work has explored non-transitivity in two-player zero-sum games within multi-agent reinforcement learning. Balduzzi et al. (2019) characterize agent interactions through convex polytopes, using their dimensionality to decompose transitive and cyclic components. Czarnecki et al. (2020) demonstrate that real-world strategy spaces exhibit a spinning top distribution, where non-transitivity peaks at middling performance levels but diminishes at either lower or higher levels. Given the presence of non-transitivity, evaluating a strategy based on its performance against a single opponent does not reliably reflect its true capability. Therefore, previous achievements in complex games such as StarCraft (Vinyals et al., 2019) and Dota 2 (OpenAI et al., 2019) employ population-based self-play training and evaluate agents through tournament-style competitions against diverse opponents. Mirroring the population-based evaluation paradigm that succeeded in non-transitive games, we adopt tournament-based comparisons in LLM-as-a-Judge frameworks to mitigate ranking instability induced by non-transitivity.

## 3. Methods

### 3.1. Measuring Non-Transitivity in Pairwise Comparisons

We employ an LLM, denoted as $m_J$, to conduct pairwise comparisons between models $m_A$ and $m_B$. The objective is to determine which of the two outputs, $o_A^{(i)}$ or $o_B^{(i)}$, better follows a given instruction $I_i$. To facilitate the comparison, each model output is assigned a unique token identifier. The antisymmetric judge function $\phi(o_A^{(i)}, o_B^{(i)} \mid m_J, I_i)$ evaluates pairs of outputs from models and determines the probability of favoring $o_A^{(i)}$ as the win rate by applying a softmax operation to the log probabilities of the corresponding model tokens. The preference of $m_A$ over $m_B$ is then quantified by taking the expected value of the judge function:

$$J(m_A \succ m_B \mid I_i) = \mathbb{E}\left[\phi(o_A^{(i)}, o_B^{(i)} \mid m_J, I_i)\right]. \quad (1)$$

**Hard Non-Transitive Cases.** To quantify non-transitivity among a triplet of models $(m_A, m_B, m_C)$, we first com-

pute the Percentage of Non-Transitive cases (PNT) over the instruction set $\mathcal{I}$, defined as:

$$\text{PNT} = \frac{1}{|\mathcal{I}|} \sum_{I_i \in \mathcal{I}} \mathbb{1}_{\text{non-trans.}}(m_A, m_B, m_C \mid m_J, I_i), \quad (2)$$

where the indicator function $\mathbb{1}_{\text{non-trans.}}$ returns 1 when the judge's preferences violate logical transitivity, and 0 otherwise. See Appendix B.1 for the complete set of conditions.

However, this metric demonstrates a limitation in sensitivity: given $J(m_A \succ m_B \mid I) = 1$ and $J(m_B \succ m_C \mid I) = 1$, it classifies both $J(m_A \succ m_C \mid I) = 0$ and $J(m_A \succ m_C \mid I) = 0.49$ as non-transitive, despite the latter exhibiting substantially stronger transitivity tendency as it is closer to the transitive threshold. Such insensitivity to transitional values near the decision boundary undermines the metric's capacity to capture nuanced deviations from ideal transitivity.

**Soft Transitivity Deviation.** To address this limitation, we propose Soft Non-Transitivity Deviation (SNTD) to measure the *degree* of non-transitivity for a single instruction with a triplet of models, defined as:

$$\text{SNTD}(m_A, m_B, m_C \mid I_i) =$$

$$\frac{1}{3} \times \mathbb{E}\left[\text{JSD}\left(\phi(o_A^{(i)}, o_B^{(i)} \mid m_J, I_i)\|\hat{\phi}(o_A^{(i)}, o_B^{(i)} \mid m_J, I_i)\right)\right.$$

$$+ \text{JSD}\left(\phi(o_B^{(i)}, o_C^{(i)} \mid m_J, I_i)\|\hat{\phi}(o_B^{(i)}, o_C^{(i)} \mid m_J, I_i)\right)$$

$$\left. + \text{JSD}\left(\phi(o_A^{(i)}, o_C^{(i)} \mid m_J, I_i)\|\hat{\phi}(o_A^{(i)}, o_C^{(i)} \mid m_J, I_i)\right)\right],$$
$$(3)$$

where the Jensen–Shannon divergence (JSD) quantifies the discrepancy between observed win rates $\phi$ and estimated win rates $\hat{\phi}$ under transitivity assumptions, as defined below.

**Estimated Win Rate.** We denote the latent quality of the outputs from models $m_A$, $m_B$, and $m_C$ on instruction $I_i$ as $\gamma_A^{(i)}$, $\gamma_B^{(i)}$, and $\gamma_C^{(i)}$, respectively. Given empirical observations $\phi$, Bradley-Terry model estimate the quality gap as:

$$s_{AB}^{(i)} = \gamma_A^{(i)} - \gamma_B^{(i)} = \ln\left(\frac{\phi(o_A^{(i)}, o_B^{(i)} \mid m_J, I_i)}{1 - \phi(o_A^{(i)}, o_B^{(i)} \mid m_J, I_i)}\right). \quad (4)$$

Based on that, we can estimate the expected win rate $\hat{\phi}$ under transitivity between any two models from a triplet $(m_A, m_B, m_C)$ by utilizing the observed win rates between the other two pairs as (See Appendix B.4 for the derivation):

$$\hat{\phi}(o_A^{(i)}, o_B^{(i)} \mid m_J, I_i) = \frac{1}{1 + e^{-(s_{AC}^{(i)} - s_{BC}^{(i)})}}. \quad (5)$$

### 3.2. Measuring Model Performance

In this section, we define metrics to quantify and rank model performance given a model pool $\mathcal{M}$, instruction dataset $\mathcal{I}$,

and judge $m_\text{J}$.

**Win Rate Against Baseline.** Through currying the judge function with a fixed baseline model $m_\text{base}$, we define the win rate against the baseline model as a rating function:

$$\mathcal{R}_\text{base}(\cdot) = \frac{1}{|\mathcal{I}|} \sum_{I_i \in \mathcal{I}} \mathbb{E}\left[ \phi(\cdot, o_{m_\text{base}}^{(i)} \mid m_\text{J}, I_i) \right]. \quad (6)$$

**Bradley-Terry Coefficients.** Given a series of pairwise comparisons, we employ the Bradley-Terry (BT) model to convert comparison outcomes into coefficients $\beta_i \in \mathbb{R}$ that quantify the strength of model $m_i$. The optimal BT coefficients $\hat{\boldsymbol{\beta}}$ are estimated by maximizing the likelihood:

$$\hat{\boldsymbol{\beta}} = \arg\max_{\boldsymbol{\beta}} \sum_i \sum_{j \neq i} \left[ W_{i,j} \cdot \ln\left( \frac{1}{1 + e^{(\beta_j - \beta_i)}} \right) \right], \quad (7)$$

where $W_{i,j}$ represents the number of times model $i$ wins against model $j$. Rather than using discrete labels $\{0, 1\}$ to count victories, we utilize the judge's preferences as soft labels, defining $W_{i,j} = \sum_{I_k \in \mathcal{I}} J(m_i \succ m_j \mid I_k)$, which yields more accurate estimations (See Appendix D).

**Elo Rating.** To establish a standardized measure of model performance, we convert Bradley-Terry coefficients to Elo ratings (Elo, 1966) by setting $\xi_i = 400 \log_{10} \beta_i$. Under this system, the probability of model $m_i$ winning against model $m_j$ is expressed as:

$$P(m_i \succ m_j) = \frac{1}{1 + 10^{(\xi_j - \xi_i)/400}}. \quad (8)$$

### 3.3. Tournament-Based Ranking

We formalize the LLM-as-a-Judge evaluation as a multi-player game framework, where evaluated language models act as players. Each player's strategy space is defined by its response generation approach under given instructions. When the judge exhibits non-transitive evaluation behavior, model assessment through fixed-opponent comparisons cannot provide reliable rankings, leading us to characterize this evaluation framework as a non-transitive game.

**Round-Robin Tournament.** Tournament-based competition with diverse opponents has been established as an effective approach for performance evaluation in non-transitive games (OpenAI et al., 2019; Vinyals et al., 2019), as it enables robust assessment of relative capabilities while mitigating the impact of non-transitivity. Based on this insight, we propose a round-robin tournament structure where each model engages in pairwise evaluation against every other model in the pool, with evaluations conducted by judge $m_\text{J}$ over instruction set $\mathcal{I}$. This method enables comprehensive model evaluation through comparisons against a diverse population of models rather than relying on a fixed perspective for assessment. We subsequently apply the Bradley-Terry

model to comparison outcomes to assign scores, which are then converted into Elo scores for the final ranking.

**Swiss-Wise Iterative Matchmaking Tournament.** While round-robin evaluation yields reliable rankings, it presents significant computational challenges at scale. Incorporating a new model into a leaderboard of size $M$ necessitates $M$ model-level comparisons compared to a single comparison in baseline-fixed frameworks. To address this computational bottleneck, we propose the **S**wiss-**W**ise **I**terative **M**atchmaking (SWIM) tournament (Algorithm 1), drawing inspiration from binary search and Swiss-system tournaments. Our approach dynamically adjusts matchmaking based on Bradley-Terry coefficients, focusing comparisons near model capability boundaries in a logarithmic manner, thereby reducing the number of comparisons to $\lceil \log_2(M) \rceil$.

### 3.4. Evaluation Setup

**Datasets.** We use the AlpacaEval dataset (Li et al., 2023), which includes a wide variety of instruction types, such as information search tasks and coding problems.

**Participating models.** We evaluate 20 models that appear on both the AlpacaEval and Chatbot Arena[1] leaderboards (see Appendix A.1 for details).

**Scenarios.** We denote significant performance advantages with $\gg$ and marginal advantages with $\approx$. Based on relative model performance, we classify model triplets $(m_A, m_B, m_C)$ into four categories:

1. Lead & Lead (**LL**): $m_A \gg m_B$ and $m_B \gg m_C$.

2. Lead & Margin (**LM**): $m_A \gg m_B$ and $m_B \approx m_C$.

3. Margin & Lead (**ML**): $m_A \approx m_B$ and $m_B \gg m_C$.

4. Margin & Margin (**MM**): $m_A \approx m_B$ and $m_B \approx m_C$.

For each scenario, we select representative model triplets based on the win rates of participating models from the AlpacaEval leaderboard (see Appendix A.2 for details).

**Judge models.** For consistency with AlpacaEval, we maintain the judge configuration and prompt templates. We examine non-transitivity in judgments using two models: GPT-4-Turbo[2] and GPT-3.5-Turbo (OpenAI et al., 2023), both with the temperature set to 0. The detailed prompt is provided in Appendix G.1.

**Position Switching.** LLMs are known to exhibit biases and inconsistencies based on the order of outputs presented in the prompt (Zheng et al., 2023; Pezeshkpour & Hruschka, 2024; Raina et al., 2024). To mitigate this bias, we employ

---

[1]Refer to Fully Style-Controlled Chatbot Arena Leaderboard (2024/09/15)

[2]GPT-4-Turbo refers to `gpt-4-1106-preview` in the win rate matrix to avoid ambiguity.

*Table 1.* We measure non-transitivity in four scenarios, evaluated by GPT-4-Turbo and GPT-3.5-Turbo. Orange cells indicate maximum PNT/SNTD values (highest non-transitivity); blue cells indicate minimum PNT/SNTD values (highest transitivity). When using GPT-4-Turbo as the judge, more non-transitivity can be observed as evaluated model performance becomes more similar and the highest non-transitivity occurs when the performances of all three models are similar; however, GPT-3.5-Turbo does not exhibit this pattern.

| Scenarios | Models | GPT-4-Turbo | | GPT-3.5-Turbo | |
|---|---|---|---|---|---|
| | | PNT | SNTD | PNT | SNTD |
| **LL** $m_A \gg m_B$ $m_B \gg m_C$ | $m_A = $ gpt-4o-2024-05-13 $m_B = $ Qwen1.5-72B-Chat $m_C = $ Mistral-7B-Instruct-v0.2 | **3.98** | **0.1121** | 21.37 | **0.2654** |
| **LM** $m_A \gg m_B$ $m_B \approx m_C$ | $m_A = $ gpt-4o-2024-05-13 $m_B = $ Qwen1.5-72B-Chat $m_C = $ claude-3-sonnet-20240229 | 5.96 | 0.1336 | 22.48 | **0.2586** |
| **ML** $m_A \approx m_B$ $m_B \gg m_C$ | $m_A = $ Yi-34B-Chat $m_B = $ Qwen1.5-72B-Chat $m_C = $ Mistral-7B-Instruct-v0.2 | **3.98** | 0.1215 | **22.86** | 0.2625 |
| **MM** $m_A \approx m_B$ $m_B \approx m_C$ | $m_A = $ Qwen1.5-72B-Chat $m_B = $ claude-3-sonnet-20240229 $m_C = $ gpt-4-0314 | **8.45** | **0.1431** | **20.87** | 0.2629 |

position switching, where each comparison is evaluated with responses in both orders. The final preference score is calculated as the mean of these balanced evaluations. To reduce the impact of API randomness, we invoke the judge function twice for each order configuration.

## 4. Non-Transitive Judge Preferences

In this section, we investigate the judge's non-transitive behaviors and analyze their underlying mechanisms.

### 4.1. Increased Non-Transitivity with Similar Model

As shown in Table 1, non-transitivity emerges across all four scenarios when GPT-4-Turbo serves as the judge. Both PNT and SNTD generally increase as the performance gap between model pairs $(m_A, m_B)$ or $(m_B, m_C)$ narrows. Notably, while scenarios **LL** and **ML** have identical PNT scores, scenario **ML** exhibits a higher SNTD value, indicating more non-transitivity. This discrepancy highlights the limitation of the PNT—it fails to capture the continuous nature of judge preferences in assessing non-transitivity. Notably, we observe similar trends across other judges and datasets, confirming the generality of the finding (See Appendix B.2).

**Weaker Judge is More Non-Transitive.** Replicating our evaluation with GPT-3.5-Turbo as the judge reveals an intriguing pattern (Table 1): both PNT and SNTD values are consistently higher than those observed with GPT-4-Turbo and remain relatively stable across all scenarios, suggesting a persistent and substantial level of non-transitivity.

Previous studies have demonstrated that GPT-4-Turbo pos-

sesses stronger reasoning capabilities and exhibits significantly less bias compared to GPT-3.5-Turbo (Zheng et al., 2023). We hypothesize that the strong non-transitivity observed with GPT-3.5-Turbo stems from its inability to distinguish the quality differences among outputs, as it is generally considered to have weaker instruction-following abilities than most participating models (Chiang et al., 2024; Lin et al., 2025; Li et al., 2023; White et al., 2025). This inability leads to preferences driven by bias predominantly, which is empirically validated in Section 4.3.

### 4.2. Aggregate Non-Transitivity

We use $J(m_A \succ m_B) = \frac{1}{|\mathcal{I}|} \sum_{I_i \in \mathcal{I}} J(m_A \succ m_B \mid I_i)$ to denote the averaged pairwise preference, representing the model-level win rate between $m_A$ and $m_B$. We subsequently perform pairwise comparisons across all models and present the win rate matrix in Figure 1 with GPT-4-Turbo as the judge to assess whether instance-level non-transitivity extends to the model-level.

**Hard Non-Transitivity at Model Level is Mild.** Surprisingly, we detect no instances of hard non-transitivity (e.g., $m_a \succ m_b$, $m_b \succ m_c$, and $m_a \prec m_c$) at the model level, which we partially attribute to the effectiveness of calibration and randomness mitigation techniques. When implementing a more aggressive approach—where positions are randomly assigned for each evaluation, reducing the process to a single call—we observe occurrences of hard non-transitivity (see Appendix C.2). Nevertheless, model-level non-transitive cases remain notably rare. We hypothesize that this scarcity stems primarily from the low proportion of non-transitive evaluations when using GPT-4-Turbo as the judge. Given the sparsity of non-transitive comparisons,

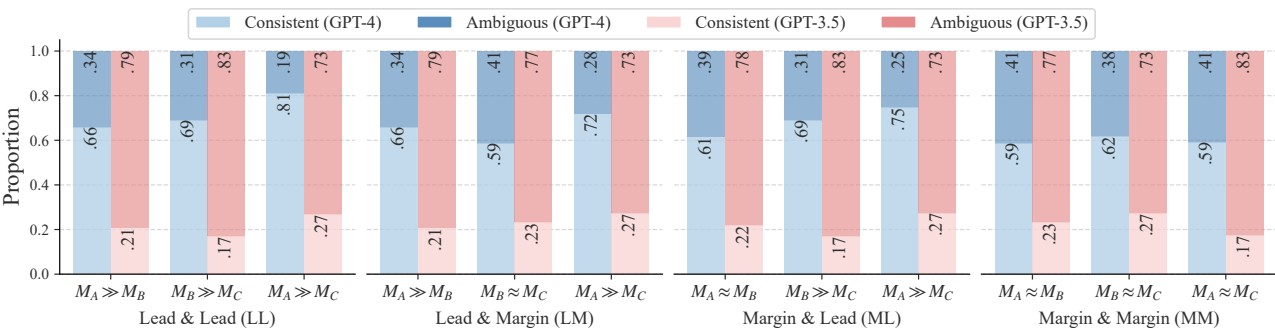

*Figure 3.* **Larger performance gaps lead to more consistent preferences.** We quantify the proportion of consistent preferences of GPT-4-Turbo and GPT-3.5-Turbo across four scenarios differentiated by relative model performance, where $\gg$ denotes substantial performance advantages and $\approx$ indicates marginal differences.

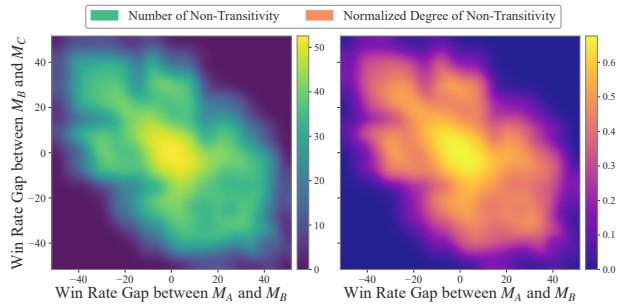

*Figure 4.* **Non-transitivity becomes more pronounced as the model performance gap approaches the origin.** We find that both PNT and SNTD peak near the origin when GPT-4-Turbo serves as the judge.

their aggregated effect is likely overwhelmed by the predominance of transitive evaluations, thus preventing the emergence of observable non-transitivity at the model level.

Despite this, notable instances of soft non-transitivity remain evident, leading to inconsistent ranking as shown by an example in Figure 1. Specifically, while GPT-4-Turbo achieves a win rate of 0.50 against GPT-4o, and GPT-4o wins against Claude-3-Opus with a rate of 0.68, transitivity would predict a win rate of 0.68 for GPT-4-Turbo against Claude-3-Opus. However, the observed rate of 0.72 reveals a subtle violation of transitivity at the model level.

**Limitations of the Baseline-Fixed Framework.** We further quantify the sensitivity of baseline-fixed frameworks. For each participating model $m$, we apply the rating function $\mathcal{R}_m(\cdot)$ to generate rankings, resulting 20 distinct ranking lists. We find that only 20% of models maintain consistent rank positions across all rankings. Moreover, when comparing any pair of ranking lists, only 61% of models preserve their rank positions on average. These findings demonstrate that rankings are highly sensitive to the choice of baseline, indicating that baseline-fixed frameworks produce inconsistent and unreliable model evaluations.

**Influence of Model Performance Difference.** We further investigate the relationship between non-transitivity and the performance gap among model triplets within all participating models. For each triplet, we define the x-axis as the win rate difference between models $m_A$ and $m_B$ from the AlpacaEval leaderboard and the y-axis as the difference between $m_B$ and $m_C$. The computed PNT and SNTD values, visualized in Figure 4, demonstrate that non-transitivity intensifies as the win rate differences between both model pairs decrease. Both metrics peak near the origin, indicating that non-transitivity is most pronounced when comparing models of similar capabilities (See Appendix B.5 for implementation details).

### 4.3. Non-Transitivity is Jointly Influenced by Position Bias and Judge's Inherent Reasoning Abilities

**Position Bias in Judge Preferences.** During the evaluation, we observe that both judges exhibit position bias. Specifically, when evaluating two models on a given instruction, we define a preference as consistent if the judge's preference maintains its relationship to 0.5 (either consistently above or below) with position switching. We report the proportion of consistent preferences in each scenario, using GPT-4-Turbo and GPT-3.5-Turbo as judges (Figure 3).

In all scenarios except **MM**, both judges show the highest preference consistency when comparing $m_A$ and $m_C$, attributable to the substantial performance gap. A potential explanation is that AlpacaEval may have limited discriminative ability when evaluating models with similar capabilities, meaning the presumed performance gap does not hold. Moreover, GPT-3.5-Turbo shows a markedly lower preference consistency than GPT-4-Turbo, indicating that its evaluations are primarily driven by position bias rather than comparing output qualities.

**Factors of Non-Transitivity.** We further categorize instructions into two groups: ambiguous and consistent. An instruction is considered consistent only when the preferences

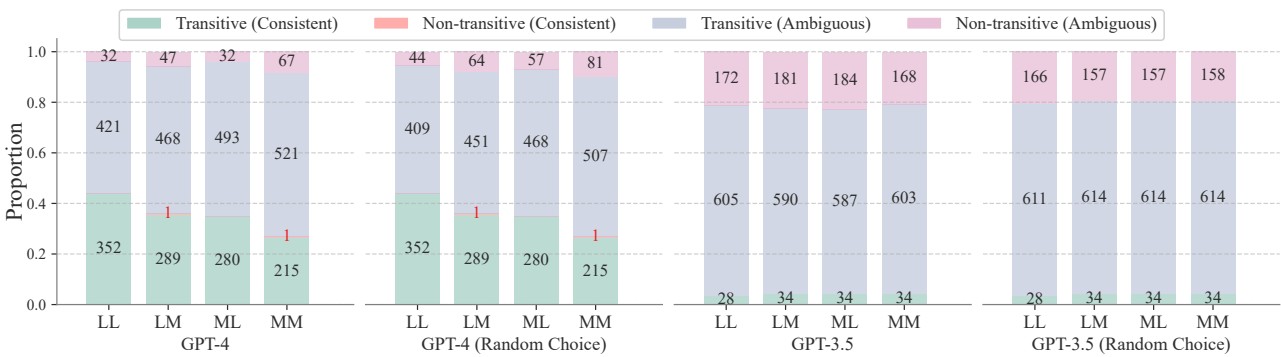

*Figure 5.* Proportion of (non-)transitive instructions across all scenarios, as evaluated by GPT-4-Turbo and GPT-3.5-Turbo. When evaluating model triplets with GPT-3.5-Turbo as judge, over 96% of instructions exhibit position bias effects. In contrast, GPT-4-Turbo demonstrates substantially higher evaluation consistency. Our analysis reveals that position switching provides more effective bias mitigation than random assignment for less position-biased judges.

between $(m_A, m_B)$, $(m_B, m_C)$, and $(m_A, m_C)$ are all consistent, implying that all comparisons are not influenced by position bias. Otherwise, the instruction is categorized as ambiguous, as at least one of the comparisons is affected by position bias. We report the proportion of non-transitive cases in Figure 5. We find that ambiguous instruction exhibits significantly higher non-transitivity rates compared to consistent instructions, suggesting position bias is indeed a contributing factor. Furthermore, when using GPT-3.5-Turbo as the judge, the proportion of ambiguous instructions exceeds 96%, validating that it exhibits a much stronger position bias than GPT-4-Turbo.

Interestingly, we find non-transitivity still occurs within consistent instructions, with GPT-4-Turbo serving as the judge, indicating that position bias is not the sole cause of non-transitivity. Therefore, we argue that non-transitivity arises from two primary factors. The first is the inherent reasoning capability of the model, which is non-transitive due to the judge's latent comparison criteria. When the quality of the outputs is similar, the judge may display preferences akin to a rock-paper-scissors dynamic. The second factor is the position bias, which affects the judge's preferences. These two factors interact and compound the occurrence of non-transitivity.

**Stronger Position Bias Increases Non-Transitivity.** To investigate the impact of position bias, we introduce Position Difference (PD). Given an instruction $I_i$ and a model triplet $(m_A, m_B, m_C)$, we define this measure as $\text{PD}(m_A, m_B, I_i) + \text{PD}(m_B, m_C, I_i) + \text{PD}(m_A, m_C, I_i)$, ranging from 0 to 3, where $\text{PD}(m_A, m_B, I_i)$ is defined as $\left| \mathbb{E}[\phi(o_A^{(i)}, o_B^{(i)} \mid m_J, I_i)] - \mathbb{E}[\phi(o_B^{(i)}, o_A^{(i)} \mid m_J, I_i)] \right|$. Using GPT-4-Turbo as the judge, we evaluate all triplet permutations and partition PD values into bins. As shown in Figure 6-Left, the proportion of non-transitive cases increases with PD, demonstrating a strong positive correlation.

**Usefulness of Position Switching.** Instead of using position switching, we repeat the experiment by randomly assigning the positions of the outputs in the prompt (Figure 5). Since all preferences in the consistent instruction are consistent, the proportion of non-transitive cases remains unchanged. However, for ambiguous instructions, we observe divergent effects: GPT-4-Turbo exhibits a significant increase in non-transitivity, while GPT-3.5 shows a slight decrease.

The distributions of judge preference (see Appendix B.6) show distinct evaluation patterns between judges. When mitigating GPT-3.5-Turbo's position bias through position switching, the model tends to generate more uncertain outcomes (averaged preference $\approx 0.5$). In contrast, GPT-4-Turbo exhibits different characteristics: while position switching occasionally introduces uncertainty, its debiased preferences generally maintain clear output distinctions. This finding suggests that position switching can reduce non-transitivity for stronger judges that are less affected by position bias, with reductions ranging from 17% to 44%. However, for weaker judges that are more susceptible to position bias, it may have the opposite effect.

**Prompting Strategies to Mitigate Non-transitivity.** We explore various prompting strategies to address non-transitivity in model judgments. Our analysis focuses on Scenario **MM**, where the capabilities of the compared models are closely matched, making it easier to observe both non-transitive behaviors and the effects of different prompts. Our findings show that providing judges with a structured evaluation checklist (Cook et al., 2024) would marginally reduce non-transitive cases. Interestingly, while incorporating Chain-of-Thought reasoning (Wei et al., 2022) helps mitigate position bias, it also leads to a higher incidence of non-transitive preferences. Moreover, allowing the judge to declare ties not only increases position bias but also further amplifies non-transitivity. See Appendix C.3 for detailed results.

*Table 2.* Correlation comparison between the round-robin-based framework and AlpacaEval, with and without length control (LC).

| Method | Spearman Correlation | | | Kendall Correlation | | |
|---|---|---|---|---|---|---|
| | w/o. LC | w. LC | Δ | w/o.LC | w. LC | Δ |
| *AlpacaEval 2.0* | *81.4%* | *95.0%* | *+13.6%* | *63.2%* | *82.1%* | *+18.9%* |
| **Round-Robin** | *85.4%* | *96.4%* | *+10.0%* | *68.4%* | *86.3%* | *+17.9%* |
| Δ | **+4.0%** | **+1.4%** | | **+5.2%** | **+4.2%** | |

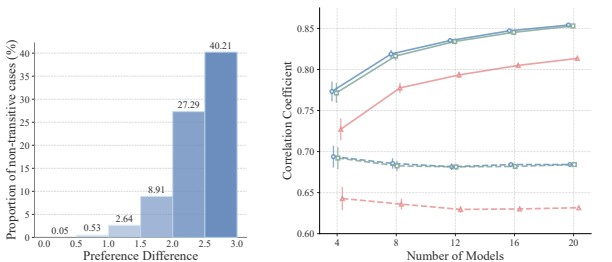

*Figure 6.* (Left) Non-transitivity strongly correlates with position difference. (Right) **Both round-robin and SWIM tournaments achieve nearly identical performance, consistently outperforming AlpacaEval in all cases.** We compare the performance between tournament-based ranking and AlpacaEval leaderboard across different numbers of participating models. For each model count, we randomly sample models and conduct 2000 trials, reporting the mean correlation with a 95% confidence interval.

## 5. Results of Tournament-Based Ranking

We conduct a round-robin tournament to obtain pairwise comparisons and apply the Bradley-Terry model to compute ratings, which are then converted to Elo scores. The resulting Elo scores and rankings for all 20 evaluated models are presented in Table 9 in the Appendix.

To assess the effectiveness of our framework, we consider the human preference ranking from the Chatbot Arena as the reference. We compute the Spearman and Kendall correlations between our round-robin-based ranking and the Chatbot Arena. We also compare these correlations with those between the AlpacaEval and the Chatbot Arena. As shown in Table 2, our method achieves higher correlations, with a 4% increase in Spearman correlation and a 5.2% increase in Kendall correlation.

**Length-Controlled Winrate.** To mitigate verbosity bias and ensure a fair comparison, we adopt the generalized linear model with the same weights as Length-Controlled AlpacaEval (Dubois et al., 2024) to derive length-controlled preferences. Using these preferences, we compute the length-controlled Bradley-Terry coefficients, which are then converted to length-controlled Elo scores. Table 2 shows that our length-controlled round-robin ranking further improves correlations, with a 1.4% increase in Spearman correlation and a 4.2% increase in Kendall correlation compared to length-controlled AlpacaEval.

**Performance of SWIM.** We demonstrate that both round-robin-based ranking and SWIM-based ranking outperform AlpacaEval, as shown in Figure 6-Right. We do not compare performance under length control, as the generalized linear model is an empirical approach that may be less interpretable, potentially affecting fairness.

## 6. Limitations and Future Work

Our study has several limitations. While AlpacaEval provides diverse instructions, it may not fully capture real-world open-ended tasks, necessitating validation of our method across broader domains. Additionally, extending our findings to judge models beyond GPT-4-Turbo and GPT-3.5-Turbo is an important direction for future work. Furthermore, while our benchmark relies on human rankings from Chatbot Arena, inherent human biases (Chen et al., 2024) may introduce non-transitivity in human preferences, fundamentally limiting the achievable alignment between automated and human evaluations.

Secondly, our focus on pairwise comparisons leaves open questions about non-transitivity in pointwise evaluations. While pointwise methods inherently avoid position bias caused by output ordering, converting these scores to pairwise comparison (A > B if score(A) > score(B)) may introduce new forms of non-transitivity, depending on the granularity and consistency of rating criteria. Future work should investigate whether such conversions preserve transitivity and identify conditions that modulate cyclic preferences.

Finally, our analysis relies on the Bradley-Terry model, which assumes transitive model-level preferences by assigning each model a global scalar score. While we do observe instance-level non-transitivity in our pairwise comparisons, these cases are relatively rare, and hard non-transitivity in the aggregated model-level preferences is mild. Therefore, we find the Bradley-Terry model sufficient for our ranking purposes. Nevertheless, we acknowledge that this implementation may not fully capture the nuanced capabilities of models. We leave this as a direction for future work, focusing on more expressive alternatives that parameterize model capabilities in a multi-dimensional space (Duan et al., 2017), which remains a promising and under-explored approach for improving the robustness of LLM-as-a-judge evaluations.

## 7. Conclusion

In this paper, we comprehensively study the impact of non-transitivity in the current LLM-based framework with pairwise settings, filling a gap in this area of research. Our findings show that non-transitivity can be observed at the instruc-

tion level during judgment and is related to the reasoning capability of the judge. The aggregation of instruction-level non-transitivity further leads to model-level non-transitivity, revealing the limitations of the baseline-fixed framework, as the rankings in this setting depend on the choice of the baseline model. Our analysis also demonstrates that position bias is a key factor in non-transitivity, with systematic position switching proving more effective than random assignment in reducing non-transitivity for stronger judges.

To address the above, we propose a baseline-free framework utilizing round-robin tournaments with Bradley-Terry model, which captures non-transitivity patterns and demonstrates better alignment with human. Recognizing the computational constraints of round-robin tournaments, which require $\mathcal{O}(nm^2)$ instruction-level comparisons for ranking $m$ models across $n$ instructions, we propose Swim tournaments. This approach achieves $\mathcal{O}(nm \log m)$ complexity through dynamic matching, substantially reducing computational cost while maintaining nearly identical performance. The code and data are available at https://github.com/yix8/llm-nontransitivity.

## Acknowledgements

We thank to the reviewers and the area chair for their constructive suggestions. We also thank to the OpenAI researcher access program for providing the OpenAI API credits used in this project. Finally, we are grateful to Akbir Khan for early comments, suggestions and advice on the project. LR is supported by the EPSRC Grant EP/S021566/1 and UCL International Scholar Award for Doctoral Training Centres.

## Impact Statement

This paper presents work whose goal is to advance the field of Machine Learning. There are many potential societal consequences of our work, none which we feel must be specifically highlighted here.

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

# A. LLM Details.

In this section, we provide detailed information about all models participating in the ranking evaluation for our experiments.

## A.1. Participating LLMs.

The experimental model set consists of 20 LLMs encompassing a range of top proprietary models, large open-source models, and small open-source models. All models are concurrently presented on the AlpacaEval leaderboard and the Fully Style-Controlled Chatbot Arena (2024/09/15). The AlpacaEval leaderboard supplies pre-generated outputs for each model on the AlpacaEval dataset, allowing us to avoid the computational costs associated with output generation and focus solely on the costs involved in the evaluation process. The Fully Style-Controlled Chatbot Arena provides human preference rankings, which we use as a reference for calculating the agreement. A detailed list of participating LLMs is presented below:

- **Proprietary models** includes four OpenAI models: `gpt-4-1106-preview`, `gpt-4o-2024-05-13`, `gpt4_0314`, `gpt-4-turbo-2024-04-09` (OpenAI et al., 2023); three Anthropic models: `claude-2`, `claude-3-opus-20240229`, `claude-3-sonnet-20240229` (Anthropic, 2023; 2024); two Mistral models: `mistral-large-2402`, `mistral-medium` (Jiang et al., 2023); one Google model: `gemini-pro` (Gemini Team Google, 2023); and one Yi model: `yi-large-preview` (01.AI, 2024).

- **Large open-source models** includes `Yi-34B-Chat` (01.AI et al., 2024), `Llama-3.1-405B-Instruct-Turbo` (Meta AI, 2024b), `Llama-3-70B-Instruct` (Meta AI, 2024a), `Qwen1.5-72B-Chat` (Qwen Team, 2024), `wizardlm-70b` (Xu et al., 2024).

- **Small open-source models** includes `Meta-Llama-3-8B-Instruct` (Meta AI, 2024a), `vicuna-13b` (Chiang et al., 2023), `Starling-LM-7B-alpha` (Zhu et al., 2024), `Mistral-7B-Instruct-v0.2` (Jiang et al., 2023).

## A.2. Selection of Representative Model Triplets across Scenarios.

For each scenario, we select representative model triplets based on the win rates of participating models (shown in parentheses) from the AlpacaEval leaderboard:

1. **LL**: GPT-4O-2024-05-13 (51.3%) as $m_A$, QWEN1.5-72B-CHAT (26.5%) as $m_B$, and MISTRAL-7B-INSTRUCT-V0.2 (14.7%) as $m_C$.

2. **LM**: GPT-4O-2024-05-13 (51.3%) as $m_A$, QWEN1.5-72B-CHAT (26.5%) as $m_B$, and CLAUDE-3-SONNET-20240229 (25.6%) as $m_C$.

3. **ML**: YI-34B-CHAT (29.7%) as $m_A$, QWEN1.5-72B-CHAT (26.5%) as $m_B$, and MISTRAL-7B-INSTRUCT-V0.2 (14.7%) as $m_C$.

4. **MM**: QWEN1.5-72B-CHAT (26.5%) as $m_A$, CLAUDE-3-SONNET-20240229 (25.6%) as $m_B$, and GPT-4-0314 (22.1%) as $m_C$.

# B. Non-Transitivity in Preference

## B.1. Conditions for Non-Transitivity

In this section, we define the conditions under which non-transitivity arises in pairwise model comparisons. Consider a triplet of models, $(m_A, m_B, m_C)$, and the corresponding pairwise comparisons on instruction $I_i$:

$$J(m_A \succ m_B \mid I_i),\ J(m_B \succ m_C \mid I_i),\ J(m_A \succ m_C \mid I_i)$$

where $J(m_x \succ m_y \mid I_i)$ denotes the preference of the judge that model $m_x$ outperforms model $m_y$ under instruction $I_i$.

Non-transitivity occurs if the results of these comparisons form any of the following patterns:

- $m_A \succ m_B,\ m_B \succ m_C,\ m_A \sim m_C$

- $m_A \succ m_B,\ m_B \succ m_C,\ m_A \prec m_C$

- $m_A \succ m_B,\ m_B \sim m_C,\ m_A \sim m_C$

- $m_A \succ m_B,\ m_B \sim m_C,\ m_A \prec m_C$

- $m_A \sim m_B,\ m_B \succ m_C,\ m_A \sim m_C$

- $m_A \sim m_B,\ m_B \succ m_C,\ m_A \prec m_C$

- $m_A \sim m_B,\ m_B \sim m_C,\ m_A \succ m_C$

- $m_A \sim m_B,\ m_B \sim m_C,\ m_A \prec m_C$

- $m_A \sim m_B,\ m_B \prec m_C,\ m_A \succ m_C$

- $m_A \sim m_B,\ m_B \prec m_C,\ m_A \sim m_C$

- $m_A \prec m_B,\ m_B \succ m_C,\ m_A \succ m_C$

- $m_A \prec m_B,\ m_B \succ m_C,\ m_A \sim m_C$

- $m_A \prec m_B,\ m_B \prec m_C,\ m_A \succ m_C$

- $m_A \prec m_B,\ m_B \prec m_C,\ m_A \sim m_C$

where $\succ$ means the left side wins against the right, $\prec$ means the left side loses to the right, and $\sim$ represents a tie between the two sides.

**Threshold Setting.** In practice, given the continuous nature of probability estimates, ties where $J(m_x \succ m_y \mid I_i) = 0.5$ occur with negligible frequency. Therefore, we introduce the following thresholds to determine the outcome of pairwise comparisons:

1. If $0.475 \leq J(m_x \succ m_y \mid I_i) \leq 0.525$, the outcome is treated as a tie ($\sim$).

2. If $J(m_x \succ m_y \mid I_i) > 0.525$, the outcome is classified as a win for $M_x$ ($\succ$).

3. If $J(m_x \succ m_y \mid I_i) < 0.475$, the outcome is classified as a loss for $M_x$ ($\prec$).

Notably, even without threshold settings, the non-transitivity patterns observed across all four scenarios remain consistent with Section 4, which is shown in Appendix B.3.

### B.2. Results Under Varying Judges and Datasets

To further assess the robustness of our findings, we evaluate the same four scenario settings on the AlpacaEval dataset using GPT-4o-mini[3] as the judge. As shown in Table 3, the results align closely with those obtained using GPT-4-Turbo: the SNTD metric confirms that non-transitivity increases as the performance gap between model pairs narrows. In addition, based on the Chatbot Arena rankings (Chiang et al., 2024), GPT-4o-mini is ranked higher than GPT-4-Turbo, suggesting that it serves as a stronger judge. Across almost all scenarios, GPT-4o-mini exhibits lower SNTD and PNT values than GPT-4-Turbo, indicating more transitive judgments. These results provide further empirical support for our claim that stronger judges tend to exhibit less non-transitivity.

To evaluate whether this pattern holds across datasets, we also conduct experiments on the Arena-Hard-Auto (Li et al., 2024) dataset, which consists of 500 high-quality prompts curated from Chatbot Arena. Due to computational constraints, we sample 200 prompts for evaluation. We utilize GPT-4-Turbo, GPT-3.5-Turbo, and GPT-4o-mini as judges under the four-scenario framework, selecting models based on their rankings in the Arena-Hard-Auto leaderboard. As shown in the Table 4, the results remain consistent with those observed on AlpacaEval: the SNTD metric confirms that non-transitivity intensifies as the performance gap narrows, particularly for stronger judges. In contrast, GPT-3.5-Turbo exhibits high non-transitivity across all scenarios, due to its inability to reliably distinguish quality differences among the outputs. This consistency suggests that the non-transitive behavior of LLM judges is robust across datasets.

---

[3]Specifically, `gpt-4o-mini-2024-07-18` is used for evaluation.

*Table 3.* We measure non-transitivity on the AlpacaEval dataset across four scenarios, evaluated by GPT-4o-mini. Orange cells indicate maximum PNT/SNTD values (highest non-transitivity); blue cells indicate minimum PNT/SNTD values (highest transitivity). Consistently, more non-transitivity can be observed as evaluated model performance becomes more similar and the highest non-transitivity occurs when the performances of all three models are similar.

| Scenarios | Models | GPT-4o-mini | |
|---|---|---|---|
| | | PNT | SNTD |
| **LL** $m_A \gg m_B$ $m_B \gg m_C$ | $m_A =$ gpt-4o-2024-05-13 $m_B =$ Qwen1.5-72B-Chat $m_C =$ Mistral-7B-Instruct-v0.2 | **3.35** | **0.1006** |
| **LM** $m_A \gg m_B$ $m_B \approx m_C$ | $m_A =$ gpt-4o-2024-05-13 $m_B =$ Qwen1.5-72B-Chat $m_C =$ claude-3-sonnet-20240229 | 3.60 | 0.1070 |
| **ML** $m_A \approx m_B$ $m_B \gg m_C$ | $m_A =$ Yi-34B-Chat $m_B =$ Qwen1.5-72B-Chat $m_C =$ Mistral-7B-Instruct-v0.2 | **3.98** | 0.1036 |
| **MM** $m_A \approx m_B$ $m_B \approx m_C$ | $m_A =$ Qwen1.5-72B-Chat $m_B =$ claude-3-sonnet-20240229 $m_C =$ gpt-4-0314 | 3.60 | **0.1173** |

*Table 4.* We measure non-transitivity on the Arena-Hard-Auto dataset across four scenarios, evaluated by GPT-4-Turbo, GPT-3.5-Turbo, and GPT-4o-mini. Orange cells indicate maximum PNT/SNTD values (highest non-transitivity); blue cells indicate minimum PNT/SNTD values (highest transitivity). We observe a similar pattern as on the AlpacaEval dataset.

| Scenarios | Models | GPT-4-Turbo | | GPT-3.5-Turbo | | GPT-4o-mini | |
|---|---|---|---|---|---|---|---|
| | | PNT | SNTD | PNT | SNTD | PNT | SNTD |
| **LL** $m_A \gg m_B$ $m_B \gg m_C$ | $m_A =$ gpt-4o-2024-05-13 $m_B =$ Qwen1.5-72B-Chat $m_C =$ Mistral-7B-Instruct | **2.00** | **0.0820** | **17.00** | 0.2071 | **1.00** | **0.0813** |
| **LM** $m_A \gg m_B$ $m_B \approx m_C$ | $m_A =$ gpt-4o-2024-05-13 $m_B =$ Mistral-Large-2402 $m_C =$ Qwen1.5-72B-Chat | 3.00 | 0.1083 | 17.50 | **0.2002** | 1.50 | 0.0880 |
| **ML** $m_A \approx m_B$ $m_B \gg m_C$ | $m_A =$ Mistral-Large-2402 $m_B =$ Qwen1.5-72B-Chat $m_C =$ Mistral-7B-Instruct | 2.50 | 0.0945 | 24.50 | **0.2370** | **5.50** | 0.1085 |
| **MM** $m_A \approx m_B$ $m_B \approx m_C$ | $m_A =$ gpt-4-0613 $m_B =$ Mistral-Large-2402 $m_C =$ Qwen1.5-72B-Chat | **5.00** | **0.1270** | **28.00** | 0.2294 | 5.00 | **0.1181** |

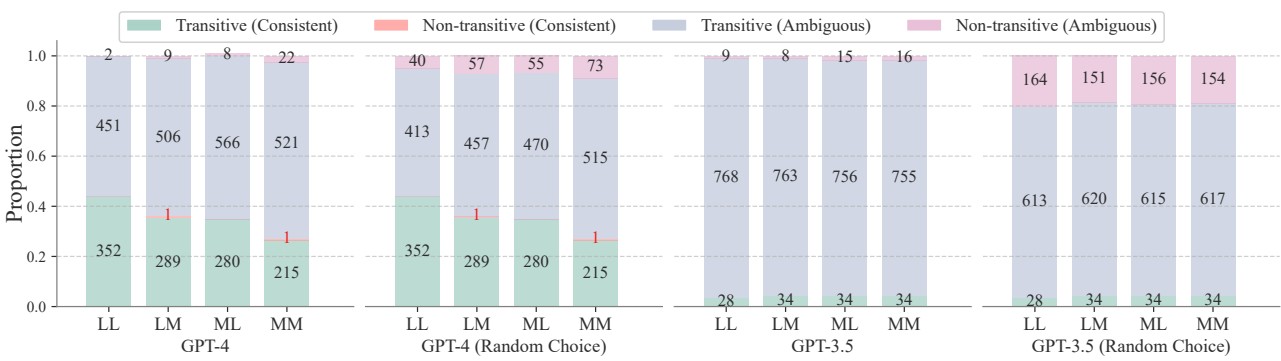

*Figure 7.* Proportion of (non-)transitive instructions across all scenarios (**without the threshold of ties**), as evaluated by GPT-4-Turbo and GPT-3.5-Turbo. When evaluating model triplets with GPT-3.5-Turbo as judge, over 96% of instructions exhibit position bias effects. In contrast, GPT-4-Turbo demonstrates substantially higher evaluation consistency.

## B.3. Results with Preferences without the Threshold of Ties

*Table 5.* We measure non-transitivity (**without the threshold of ties**) on the AlpacaEval dataset across four scenarios, evaluated by GPT-4-Turbo and GPT-3.5-Turbo. Orange cells indicate maximum PNT/SNTD values (highest non-transitivity); blue cells indicate minimum PNT/SNTD values (highest transitivity). When using GPT-4-Turbo as the judge, more non-transitivity can be observed as evaluated model performance becomes more similar and the highest non-transitivity occurs when the performances of all three models are similar; however, GPT-3.5-Turbo does not exhibit this pattern.

| Scenarios | Models | *GPT-4-Turbo* | | *GPT-3.5-Turbo* | |
|---|---|---|---|---|---|
| | | PNT | SNTD | PNT | SNTD |
| **LL** $m_A \gg m_B$ $m_B \gg m_C$ | $m_A = $ gpt-4o-2024-05-13 $m_B = $ Qwen1.5-72B-Chat $m_C = $ Mistral-7B-Instruct-v0.2 | **0.25** | **0.1121** | 1.12 | **0.2654** |
| **LM** $m_A \gg m_B$ $m_B \approx m_C$ | $m_A = $ gpt-4o-2024-05-13 $m_B = $ Qwen1.5-72B-Chat $m_C = $ claude-3-sonnet-20240229 | 1.24 | 0.1336 | **0.25** | **0.2586** |
| **ML** $m_A \approx m_B$ $m_B \gg m_C$ | $m_A = $ Yi-34B-Chat $m_B = $ Qwen1.5-72B-Chat $m_C = $ Mistral-7B-Instruct-v0.2 | 0.99 | 0.1215 | 1.86 | 0.2625 |
| **MM** $m_A \approx m_B$ $m_B \approx m_C$ | $m_A = $ Qwen1.5-72B-Chat $m_B = $ claude-3-sonnet-20240229 $m_C = $ gpt-4-0314 | **2.86** | **0.1431** | **1.99** | 0.2629 |

We observe the same pattern from the table 5 as in the main text, which is with the threshold for ties. When GPT-4-Turbo serves as the judge, both PNT and SNTD increase as the performance gap between any pair of models, $(m_A, m_B)$ or $(m_B, m_C)$, decreases. In cases where all three models exhibit similar performance, such as in scenario **MM**, the incidence of non-transitivity rises significantly. We attribute this to the increased uncertainty judges face when assessing quality differences between similar outputs. When the comparisons between $m_A$ and $m_B$, $m_B$ and $m_C$, and $m_A$ and $m_C$ are all uncertain, non-transitivity reaches its highest level. Replicating our evaluation with GPT-3.5-Turbo as judge reveals an intriguing pattern: while the PNT remains minimal across scenarios, the consistently high SNTD values indicate substantial non-transitivity. This observation motivates us to define the tie threshold, as ties can serve as an indicator of model uncertainty.

To explain the low number of hard non-transitive cases when using GPT-3.5-Turbo as the judge with position switching in Figure 5, we hypothesize that GPT-3.5-Turbo is also affected by other biases (Zhou et al., 2024a), such as verbosity bias (Saito et al., 2023) and token bias (Alzahrani et al., 2024). Since GPT-3.5-Turbo struggles to accurately assess the quality of outputs, these combined biases influence the judge's preferences. As a result, even though position switching mitigates the position bias, the averaged preference is still not determined by the actual quality of the outputs but rather by other fixed

biases in the prompt, leading to transitive preferences. This observation also motivates us to define the threshold, as it can be used to reduce the impact of other biases.

### B.4. Derivation of Expected Win Rate

The Bradley-Terry model (Bradley & Terry, 1952) provides a probabilistic framework for estimating pairwise win rates based on these latent quality scores. Specifically, the probability that model $m_A$ outperforms model $m_B$ on instruction $I_i$ is given by:

$$\phi(o_A^{(i)}, o_B^{(i)} \mid m_J, I_i) = \frac{1}{1 + e^{-(\gamma_A^{(i)} - \gamma_B^{(i)})}} = \sigma(s_{AB}^{(i)}), \tag{9}$$

where we denote $s_{AB}^{(i)} = \gamma_A^{(i)} - \gamma_B^{(i)}$ as the quality gap. Conversely, this quality gap can be calculated from empirical observations $\phi$ as:

$$s_{AB}^{(i)} = \ln\left(\frac{\phi(o_A^{(i)}, o_B^{(i)} \mid m_J, I_i)}{1 - \phi(o_A^{(i)}, o_B^{(i)} \mid m_J, I_i)}\right). \tag{10}$$

Based on that, we can estimate the expected win rate $\hat{\phi}$ under transitivity between any two models from a triplet $(m_A, m_B, m_C)$ by utilizing the observed win rates between the other two pairs. For instance, to estimate the win rate for model $m_A$ beating model $m_B$ on instruction $I_i$ without direct observations, we assume that the observed win rates for the remaining pairs reflect true performance differences and compute the estimated win rate as:

$$\hat{\phi}(o_A^{(i)}, o_B^{(i)} \mid m_J, I_i) = \frac{1}{1 + e^{-\left((\gamma_A^{(i)} - \gamma_C^{(i)}) - (\gamma_B^{(i)} - \gamma_C^{(i)})\right)}} = \frac{1}{1 + e^{-(s_{AC}^{(i)} - s_{BC}^{(i)})}}. \tag{11}$$

### B.5. Heatmap Implementation

In this experiment, we aim to investigate the relationship between non-transitivity and the performance gap between two models being compared. From the pool of 20 models, we generate all possible tuples $(m_A, m_B, m_C)$ by computing $P(20, 3) = \frac{20!}{17!} = 6,840$ permutations. For each tuple, we calculate the number of hard non-transitive cases and the degree of soft non-transitivity. The results are visualized as a 2D heatmap, where the x-axis represents the performance gap between model $m_A$ and model $m_B$, measured by their win-rate difference on AlpacaEval. Similarly, the y-axis represents the win-rate difference between model $m_B$ and model $m_C$. A positive win-rate difference indicates that the former model performs better, whereas a negative difference suggests that the latter outperforms the former.

According to the AlpacaEval leaderboard, `yi-large-preview` achieves the highest relative win rate of 57.5%, while `vicuna-13b` records the lowest at 5.8%. This establishes a win rate differential range of [-51.7%, +51.7%], which we partition into a 35 × 35 grid. For each grid cell, we compute the mean number of PNT and SNTD across all possible model triplet permutations. We apply a Gaussian filter ($\sigma = 1$) to reduce noise in the resulting data, and then perform quadratic interpolation to generate the final heatmap.

### B.6. Preference Distributions of Judge

All scenario assumes that $m_A$ outperforms $m_B$, $m_B$ outperforms $m_C$, and $m_A$ outperforms $m_C$. Consequently, we expect the judge's preference distribution to exhibit a heavy-tailed pattern concentrated around 1. In scenario **LL**, because the models differ significantly in performance, the judge should tend to select the superior output. However, under the random assignment setting, GPT-3.5-Turbo exhibits a U-shaped distribution across all scenarios (Figure 8), validating that it fails to distinguish response quality and is instead primarily driven by position bias. As a result, after applying position switching, its preference distribution changes significantly, forming a sharp peak at 0.5 while rapidly decaying away from it, leading to a large number of ties.

By contrast, GPT-4-Turbo's distributions vary across scenarios (Figure 9). In scenario **LL**, where $m_A$, $m_B$, and $m_C$ have large performance gaps, the distribution precisely follows a heavy-tailed pattern concentrated at 1, indicating that when GPT-4-Turbo perceives a substantial quality difference, it strongly favors the superior response. In **LM** and **ML** scenarios, where one model pair has a clear performance gap while the other is closer in quality, increased uncertainty arises when evaluating the latter, causing the tail to shift towards 0. In **MM**, GPT-4-Turbo also exhibits a U-shaped distribution. However, unlike GPT-3.5-Turbo, it retains 38% of its preferences distributed across the full range from 0 to 1, demonstrating that its

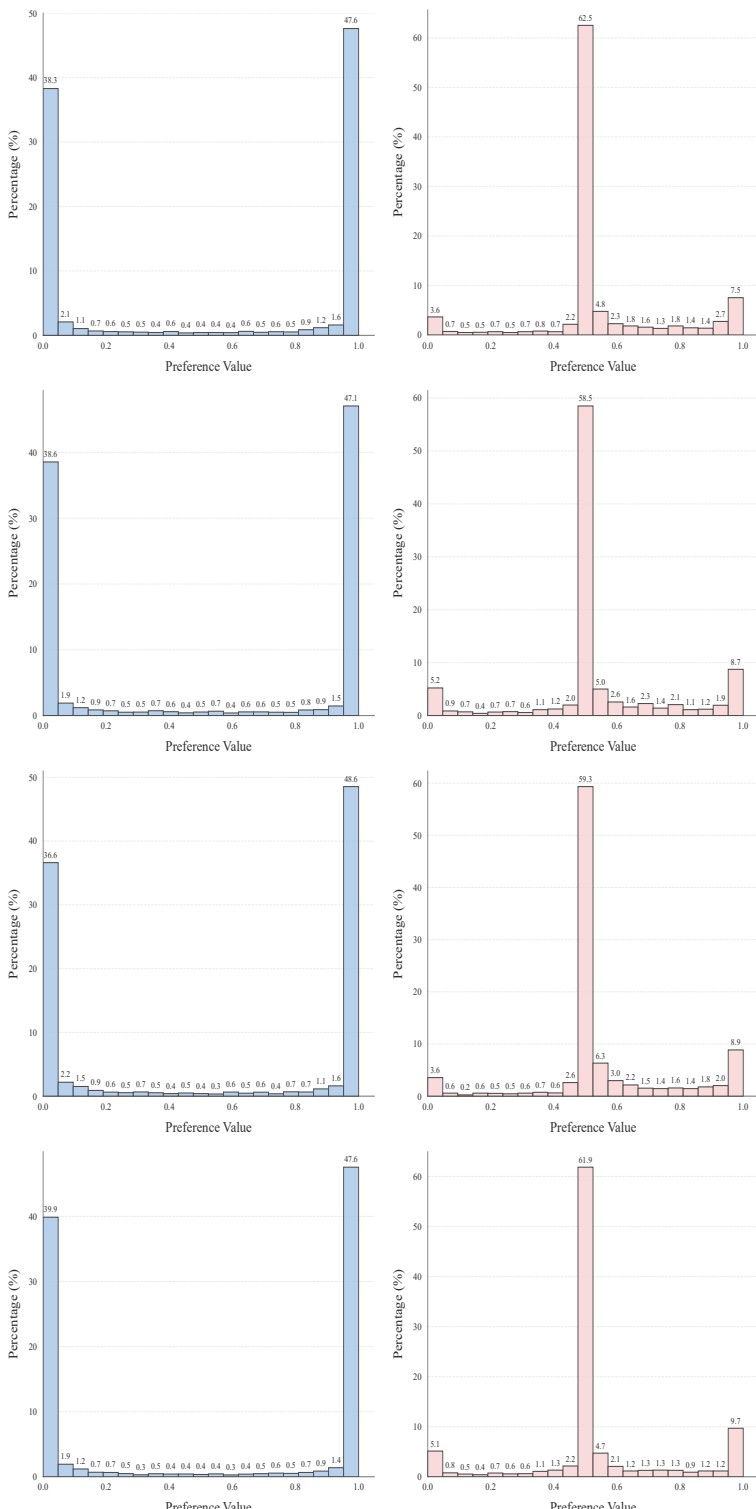

*Figure 8.* Preference distribution of GPT-3.5-Turbo across scenarios (from top to bottom: **LL**, **LM**, **LM**, **MM**). (Left) Distribution with random assignment. (Right) Distribution with position switching.

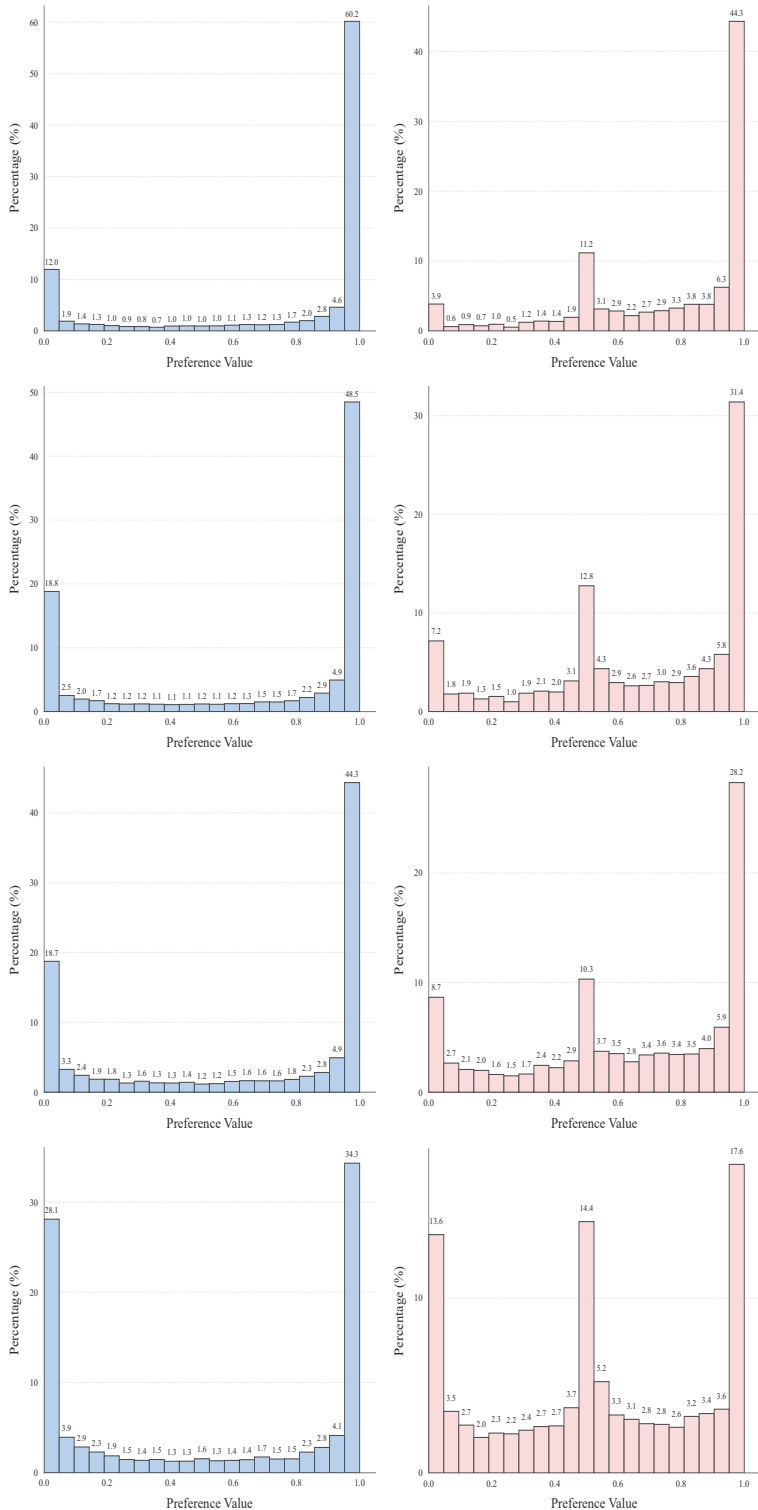

*Figure 9.* Preference distribution of GPT-4-Turbo across scenarios (from top to bottom: **LL**, **LM**, **LM**, **MM**). (Left) Distribution with random assignment. (Right) Distribution with position switching.

preferences are guided by reasoning rather than solely by position bias. Thus, position switching smooths its preference distribution while preserving a considerable proportion of decisive judgments (non-ties), reflecting that GPT-4-Turbo still distinguishes quality differences

This also explains why position switching is least effective in Scenario **MM**, reducing non-transitivity by only 17%.

## C. Additional Experimental Results

### C.1. Full Pairwise Comparison Matrix (Position Switching and Two API Calls per Order)

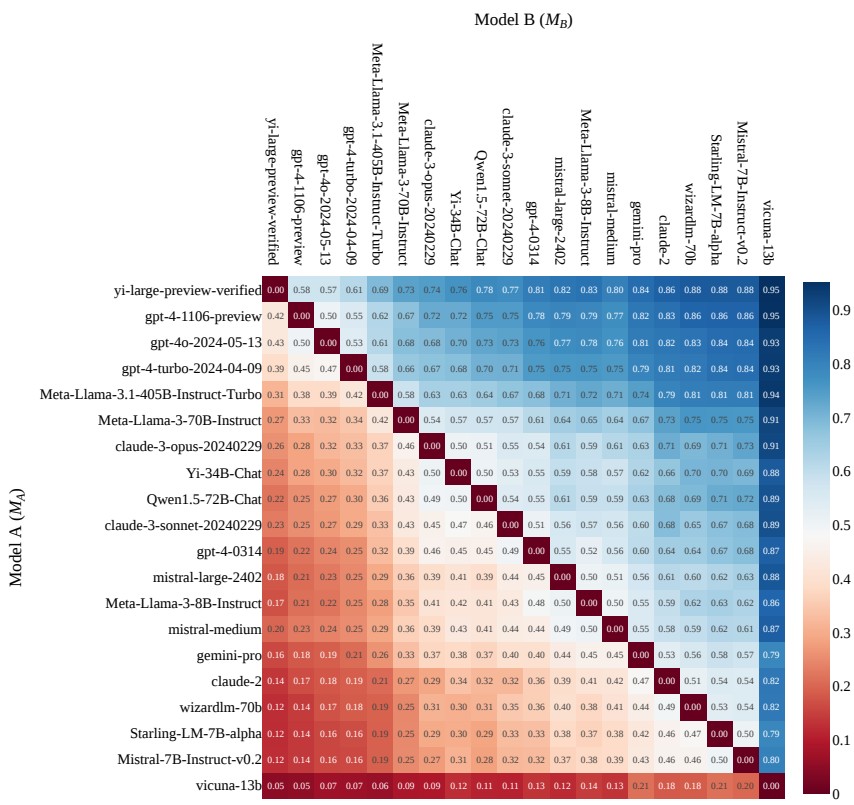

*Figure 10.* Win rate matrix for 20 models using default settings (Position Switching and Two API Calls per Order).

### C.2. Position Switching and Multiple API Calls Reduce the Occurrence of Non-transitivity at the Model Level.

We hypothesize that the absence of observed hard non-transitivity in Figure 10 is due to the use of position switching and two API calls per order, which help ensure the consistency of judgments. To validate this hypothesis, we adopt a more aggressive approach by randomly assigning positions for each evaluation, reducing the process to a single API call to mitigate position bias. However, since the preference between each model pair for a given instruction is determined by log probability rather than a binary label (0 or 1), we argue that random assignment may not fully eliminate position bias. As a result, this setup is expected to perform worse than position switching, leading to lower judgment consistency compared to the original setting.

To reduce computational costs, the judge's new preference can be interpreted as a random sample from the four API calls made in the original experiment. In other words, in this ablation experiment, the judge's preference is equivalent to selecting one random sample from the pre-computed preferences in Section 4.2.

Figure 11 presents the corresponding win rate matrix from this ablation. In contrast to Figure 10, we now observe the occurrence of a hard non-transitive case at the model level. Specifically, `Qwen1.5-72B-Chat` outperforms `Yi-34B-Chat`, and `Yi-34B-Chat` outperforms `claude-3-opus-20240229`. However, `claude-3-opus-20240229` outperforms `Qwen1.5-72B-Chat`, thus exhibiting a clear case of non-transitivity.

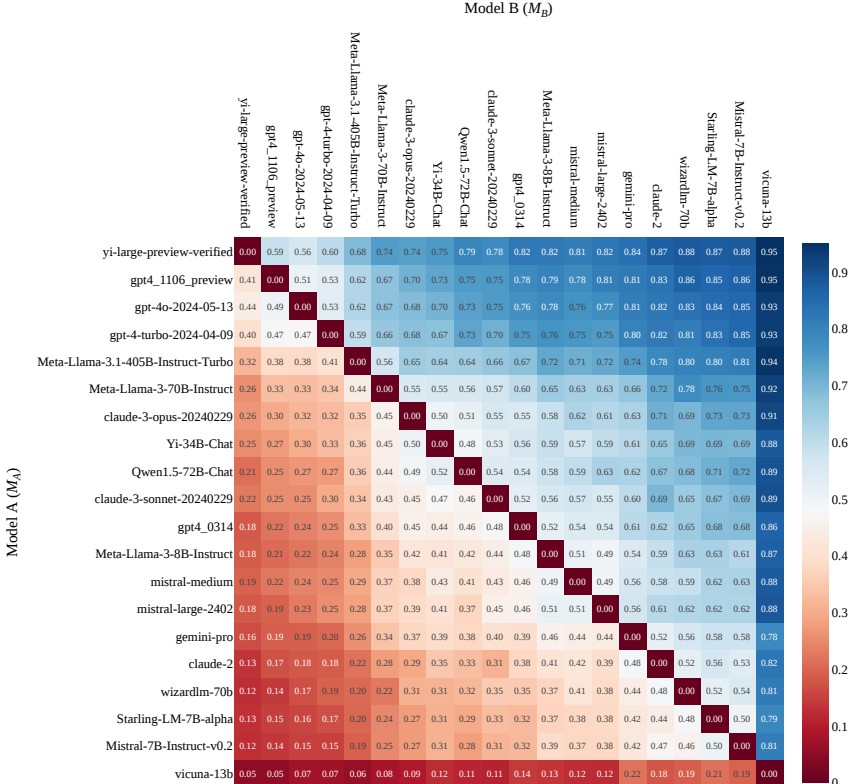

*Figure 11.* Win rate matrix for 20 models using ablated settings (random assignment and a single API call per order). Hard non-transitivity is observed compared to Figure 1. For instance, `Qwen1.5-72B-Chat` outperforms `Yi-34B-Chat`, and `Yi-34B-Chat` outperforms `claude-3-opus-20240229`. However, `claude-3-opus-20240229` outperforms `Qwen1.5-72B-Chat`, highlighting the presence of non-transitive relationships among the models.

To further verify that the observation of non-transitivity in the ablated setting is not merely due to randomness, we repeat this ablation experiment 50 times. We quantify the degree of soft non-transitivity in the win rate matrix in a manner similar to Equation 3, but applied at the model level. Specifically, for a set of 20 models, we first compute all possible permutations of triples $(m_A, m_B, m_C)$. For each triplet, we sequentially select two pairs of models and extract their corresponding values from the win rate matrix as ground truth. We then calculate the expected win rate for the remaining model pair and measure the associated SNTD at the model level. Finally, we average the results across all permutations to assess the overall non-transitivity in the win-rate matrix.

*Table 6.* Comparison of the degree of soft non-transitivity between the original and random assignment settings. The values represent the mean SNTD, with the standard deviation reported for the random assignment setting based on 50 independent trials.

| Experiment Setting | SNTD |
|---|---|
| Position Switching and Two API Calls per Order | $4.00 \times 10^{-4}$ |
| Random Assignment and One API Call (50 times) | $(\mathbf{5.38 \pm 0.04}) \times 10^{-4}$ |

As shown in Table 6, the degree of non-transitivity in the ablated experiment is significantly higher than in the original experiment. This finding demonstrates that by employing position switching and multiple API calls, we can improve the consistency of the judge's evaluations and thereby reduce the occurrence of non-transitivity at the model level.

## C.3. More Prompting Strategies

We evaluate six prompting strategies in Scenario **MM** to encourage the judge to exhibit more transitive preferences from a prompting perspective ($\approx\approx$). The prompt templates are provided in Appendix G.1.

1. **Direct Comparison**: Standard binary choice comparison identical to our previous experimental setup, serving as the baseline.

2. **CoT Comparison**: Requires the judge to output its reasoning through Chain-of-Thought (Wei et al., 2022) before making a decision.

3. **Direct Comparison with Checklist**: Provides a detailed evaluation checklist (Cook et al., 2024) for the judgment without explicit reasoning.

4. **CoT Comparison with Checklist**: Combines a detailed evaluation checklist with Chain-of-Thought reasoning before judgment.

5. **CoT Comparison (Tie Allowed)**: Extends the binary choice to three options by introducing the possibility of ties, while maintaining the Chain-of-Thought reasoning process.

6. **CoT Comparison with Checklist (Tie Allowed)**: Incorporates both the three-choice option and evaluation checklist while preserving Chain-of-Thought reasoning.

*Table 7.* Comparison of different prompting strategies, judged by GPT-4-Turbo. Red cells indicate the lowest consistency (most affected by position bias); green cells represent the highest consistency (least affected by position bias). Orange cells denote the highest number of non-transitive cases (greatest non-transitivity), while blue cells indicate the lowest number of non-transitive cases (greatest transitivity). The values in parentheses represent the number of non-transitive cases in consistent instructions (left) and ambiguous instructions (right).

| Method | A vs B (consist.) | B vs C (consist.) | A vs C (consist.) | # of Consistent Instr. | # of Non-trans. (w. threshold) | # of Non-trans. (w/o. threshold) |
|---|---|---|---|---|---|---|
| Direct | 473 | 496 | 476 | 217 | (1, 67) | (1, 22) |
| Direct w. Chk | 478 | 506 | 440 | 227 | (0, 64) | (0, 23) |
| CoT | 572 | 577 | 560 | 301 | (1, 152) | (1, 46) |
| CoT w. Chk | 548 | 571 | 535 | 268 | (5, 172) | (5, 47) |
| CoT w. Tie | 474 | 496 | 493 | 210 | (5, 139) | (5, 87) |
| CoT w. Chk&Tie | 466 | 479 | 456 | 181 | (10, 183) | (10, 129) |

For the checklist-based method, we first use GPT-4-Turbo to generate a checklist—a set of YES/NO questions assessing different aspects of the given instruction. The corresponding prompt is provided in Appendix G.2.

As shown in Table 7, providing the judge with a checklist slightly reduces non-transitivity. This aligns with our earlier assertion that the judge's latent comparison criteria are inherently non-transitive for closely matched models. While introducing explicit criteria helps guide the judge toward more transitive preferences, the effect remains limited, likely because the automatically generated checklists lack the granularity to capture subtle output differences.

Meanwhile, although Chain-of-Thought prompting reduces position bias and improves overall preference consistency, it increases non-transitivity for ambiguous instructions and can introduce additional non-transitive cases even in consistent instructions. Additionally, when combining CoT with a checklist, we observe more inconsistency, suggesting that CoT elicits the judge's latent reasoning criteria, which may conflict with the explicitly provided checklist. Furthermore, allowing the judge to declare ties increases non-transitivity, as the judge may opt for ties instead of identifying subtle differences between outputs.

## D. Soft Bradley-Terry Model Yields More Accurate Rankings

We explored three methods for computing $W_{i,j}$ in Equation (7). The first method, referred to as hard-BT, directly derives discrete win rates from the judge's continuous preferences. In this approach, if $J(m_i \succ m_j \mid I_k) > 0.5$, the outcome is counted as a win (1); if $J(m_i \succ m_j \mid I_k) < 0.5$, it is counted as a loss (0); and if $J(m_i \succ m_j \mid I_k) = 0.5$, it is considered a tie (0.5).

The second method, rounded-BT, incorporates a threshold to refine the win/loss definition. Specifically, if $J(m_i \succ m_j \mid I_k) > 0.525$, it is considered a win (1); if $J(m_i \succ m_j \mid I_k) < 0.475$, it is considered a loss (0); and if $J(m_i \succ m_j \mid I_k)$ falls within the range $[0.475, 0.525]$, it is treated as a tie (0.5).

The final method, soft-BT, follows the formulation presented in the main text. Instead of discretizing preferences into fixed categories, it directly uses the judge's continuous preference scores to compute $W_{i,j}$, allowing for a more nuanced representation of the relative strength between models:

$$W_{i,j} = \sum_{I_k \in \mathcal{I}} J(m_i \succ m_j \mid I_k).$$

We evaluate these methods by computing rankings from a round-robin tournament involving 20 models, using GPT-4-Turbo as the judge, and measuring their correlation with the Chatbot Arena rankings as metrics.

*Table 8.* Comparison between Round Robin based framework with Bradley-Terry model and AlpacaEval 2.0.

|  | RR + Soft-BT | RR + Hard-BT | RR + Rounded-BT |
|---|---|---|---|
| Spearman Correlation | **85.4%** | 84.4% | 84.8% |
| Kendall Correlation | **68.4%** | 66.3% | 67.4% |

Table 8 shows that soft-BT produces the most aligned ranking, demonstrating its ability to better capture the relative strength of models from continuous preferences.

## E. Swiss-Wise Iterative Matchmaking tournaments

---
**Algorithm 1** **S**wiss-**W**ise **I**terative **M**atchmaking (SWIM) tournament
---

1: **Input:** $M$ unranked models, a dataset $\mathcal{I}$ and a judge model $M_J$.
2: **Output:** An ordered ranking of all $M$ models.
3: $R \leftarrow$ empty set $\varnothing$ to store ranked models
4: $U \leftarrow$ set of all $M$ models
5: $X \leftarrow$ a random model from $U$
6: $R \leftarrow R \cup \{X\}, U \leftarrow U \setminus \{X\}$
7: **while** $U \neq \varnothing$ **do**
8:     $P \leftarrow$ a random model from $U$
9:     $U \leftarrow U \setminus \{P\}$
10:     $s \leftarrow |R|, c \leftarrow \lceil \max(\log_2(s), 1) \rceil$
11:     $X \leftarrow$ a random model from $R$
12:     $T \leftarrow R \setminus \{X\}$
13:     **for all** $I_i \in \mathcal{I}$ **do**
14:         Compute $J(m_P \succ m_X \mid I_i)$
15:     **end for**
16:     $\beta \leftarrow$ update BT coefficient for $R \cup \{P\}$
17:     **for** $j = 1$ to $c - 1$ **do**
18:         $O \leftarrow \arg\min_{O \in T} |\beta_O - \beta_P|$
19:         $T \leftarrow T \setminus \{O\}$
20:         **for all** $I_i \in \mathcal{I}$ **do**
21:             Compute $J(m_P \succ m_O \mid I_i)$
22:         **end for**
23:         $\beta \leftarrow$ update BT coefficient for $R \cup \{P\}$
24:     **end for**
25:     $R \leftarrow R \cup \{P\}$
26: **end while**

---

## F. ELO Scores

We conduct a round-robin tournament to obtain pairwise comparisons and apply the Bradley-Terry model to compute ratings, which are then converted to Elo scores.

*Table 9.* Evaluation Results of LLMs in Fully Style-Controlled Chatbot Arena, Round-Robin Tournament and AlpacaEval.

| Model Names | FSC Arena Elo | Round-Robin + BT | | AlpacaEval 2.0 | |
|---|---|---|---|---|---|
| | | Elo | LC Elo | Win Rate | LC Win Rate |
| gpt-4o-2024-05-13 | 1262 | 1325 | 1227 | 51.3% | 57.5% |
| gpt-4-turbo-2024-04-09 | 1241 | 1306 | 1217 | 46.1% | 55.0% |
| gpt-4-1106-preview | 1234 | 1337 | 1206 | 50.0% | 50.0% |
| yi-large-preview | 1204 | 1377 | 1205 | 57.5% | 51.9% |
| claude-3-opus-20240229 | 1238 | 1180 | 1156 | 29.1% | 40.5% |
| Llama-3.1-405B-Instruct-Turbo | 1250 | 1264 | 1136 | 39.1% | 39.3% |
| gpt4_0314 | 1200 | 1137 | 1117 | 22.1% | 35.3% |
| claude-3-sonnet-20240229 | 1197 | 1152 | 1110 | 25.6% | 34.9% |
| Qwen1.5-72B-Chat | 1148 | 1168 | 1108 | 26.5% | 36.6% |
| Llama-3-70B-Instruct | 1193 | 1210 | 1093 | 33.2% | 34.4% |
| mistral-large-2402 | 1158 | 1110 | 1090 | 21.4% | 32.7% |
| claude-2 | 1144 | 1043 | 1060 | 17.2% | 28.2% |
| mistral-medium | 1141 | 1109 | 1059 | 21.9% | 28.6% |
| Yi-34B-Chat | 1100 | 1169 | 1026 | 29.7% | 27.2% |
| gemini-pro | 1132 | 1074 | 1020 | 18.2% | 24.4% |
| Llama-3-8B-Instruct | 1141 | 1110 | 988 | 22.6% | 22.9% |
| wizardlm-70b | 1106 | 1036 | 964 | 14.4% | 17.6% |
| Mistral-7B-Instruct-v0.2 | 1067 | 1019 | 947 | 14.7% | 17.1% |
| Starling-LM-7B-alpha | 1083 | 1021 | 925 | 14.2% | 14.7% |
| vicuna-13b | 1060 | 800 | 800 | 6.7% | 10.5% |

*Table 10.* Ranking of LLMs based on evaluation results from the Fully Style-Controlled Chatbot Arena, Round-Robin Tournament, and AlpacaEval. The numbers in parentheses indicate changes in model rankings after applying the length-controlled debiasing technique, where ↑ denotes an increase, ↓ denotes a decrease, and – indicates no change in ranking.

| Model Names | FSC Arena Rank | Round-Robin + BT | | AlpacaEval 2.0 | |
|---|---|---|---|---|---|
| | | Rank | LC Rank | Rank | LC Rank |
| gpt-4o-2024-05-13 | 1 | 3 | 1 (2 ↑) | 2 | 1 (1 ↑) |
| gpt-4-turbo-2024-04-09 | 3 | 4 | 2 (2 ↑) | 4 | 2 (2 ↑) |
| gpt-4-1106-preview | 5 | 2 | 3 (1 ↓) | 3 | 4 (1 ↓) |
| yi-large-preview | 6 | 1 | 4 (3 ↓) | 1 | 3 (2 ↓) |
| claude-3-opus-20240229 | 4 | 7 | 5 (2 ↑) | 8 | 5 (3 ↑) |
| Llama-3.1-405B-Instruct-Turbo | 2 | 5 | 6 (1 ↓) | 5 | 6 (1 ↓) |
| gpt4_0314 | 7 | 11 | 7 (4 ↑) | 12 | 8 (4 ↑) |
| claude-3-sonnet-20240229 | 8 | 10 | 8 (2 ↑) | 10 | 9 (1 ↑) |
| Qwen1.5-72B-Chat | 11 | 9 | 9 (0 –) | 9 | 7 (2 ↑) |
| Llama-3-70B-Instruct | 9 | 6 | 10 (4 ↓) | 6 | 10 (4 ↓) |
| mistral-large-2402 | 10 | 12 | 11 (1 ↑) | 14 | 11 (3 ↑) |
| claude-2 | 12 | 16 | 12 (4 ↑) | 16 | 13 (3 ↑) |
| mistral-medium | 13 | 14 | 13 (1 ↑) | 13 | 12 (1 ↑) |
| Yi-34B-Chat | 17 | 8 | 14 (6 ↓) | 7 | 14 (7 ↓) |
| gemini-pro | 15 | 15 | 15 (0 –) | 15 | 15 (0 –) |
| Llama-3-8B-Instruct | 14 | 13 | 16 (3 ↓) | 11 | 16 (5 ↓) |
| wizardlm-70b | 16 | 17 | 17 (0 –) | 18 | 17 (1 ↑) |
| Mistral-7B-Instruct-v0.2 | 19 | 19 | 18 (1 ↑) | 17 | 18 (1 ↓) |
| Starling-LM-7B-alpha | 18 | 18 | 19 (1 ↓) | 19 | 19 (0 –) |
| vicuna-13b | 20 | 20 | 20 (0 –) | 20 | 20 (0 –) |

# G. Prompt Template.

## G.1. Judge Prompts

**Direct Comparison - Identical to AlpacaEval 2.0 (Li et al., 2023)**

```
[System Part]
You are a highly efficient assistant, who evaluates and selects the best large language
    model (LLMs) based on the quality of their responses to a given instruction. This
    process will be used to create a leaderboard reflecting the most accurate and human-
    preferred answers.

[User Part]
I require a leaderboard for various large language models. I'll provide you with prompts
    given to these models and their corresponding outputs. Your task is to assess these
    responses, and select the model that produces the best output from a human perspective
    .

## Instruction

{
    "instruction": """{instruction}""",
}

## Model Outputs

Here are the unordered outputs from the models. Each output is associated with a specific
    model, identified by a unique model identifier.

{
    {
        "model_identifier": "m",
        "output": """{output_1}"""
    },
    {
```

```
        "model_identifier": "M",
        "output": """{output_2}"""
    }
}

## Task

Evaluate the models based on the quality and relevance of their outputs, and select the
    model that generated the best output. Answer by providing the model identifier of the
    best model. We will use your output as the name of the best model, so make sure your
    output only contains one of the following model identifiers and nothing else (no
    quotes, no spaces, no new lines, ...): m or M.

## Best Model Identifier
```

**Direct Comparison with Checklist**

```
[System Part]
You are a highly efficient assistant, who evaluates and selects the best large language
    model (LLMs) based on the quality of their responses to a given instruction and the
    corresponding criteria. This process will be used to create a leaderboard reflecting
    the most accurate and human-preferred answers.

[User Part]
I require a leaderboard for various large language models. I will provide you with prompts
     given to these models and their corresponding outputs. I will also provide one
    specific evaluation checklist which contains a list of specific criteria that a good
    output should fulfill. Your task is to assess these responses to see whether they
    satisfy the requirements of the checklist and select the model that produces the best
    output from a human perspective based on the provided checklist.

## Instruction

{
    "instruction": """{instruction}""",
}

## Checklist
Here is the checklist that contains the conditions specified in the question for a good
    output. The more requirements an output meets, the better it is considered.

{
    checklist: """{checklist}""",
}

## Model Outputs

Here are the unordered outputs from the models. Each output is associated with a specific
    model, identified by a unique model identifier.

{
    {
        "model_identifier": "m",
        "output": """{output_1}"""
    },
    {
        "model_identifier": "M",
        "output": """{output_2}"""
    }
}

## Task

Evaluate the models based on the quality and relevance of their outputs, and select the
    model that generated the best output based on the checklist. Answer by providing the
```

```
    model identifier of the best model. We will use your output as the name of the best
    model, so make sure your output only contains one of the following model identifiers
    and nothing else (no quotes, no spaces, no new lines, ...): m or M.

## Best Model Identifier
```

**CoT Comparison - Identical to AlpacaEval 2.0 (Li et al., 2023)**

```
[System Part]
You are a highly efficient assistant, who evaluates and selects the best large language
    model (LLMs) based on the quality of their responses to a given instruction. This
    process will be used to create a leaderboard reflecting the most accurate and human-
    preferred answers.

[User Part]
I require a leaderboard for various large language models. I'll provide you with prompts
    given to these models and their corresponding outputs. Your task is to assess these
    responses, and select the model that produces the best output from a human perspective
    .

## Instruction

{
    "instruction": """{instruction}""",
}

## Model Outputs

Here are the unordered outputs from the models. Each output is associated with a specific
    model, identified by a unique model identifier.

{
    {
        "model_identifier": "m",
        "output": """{output_1}"""
    },
    {
        "model_identifier": "M",
        "output": """{output_2}"""
    }
}

## Task

Evaluate the models based on the quality and relevance of their outputs, and select the
    model that generated the best output. Answer by first providing a concise explanation
    and then end your answer by providing the model identifier of the best output. We will
     use the last character of your output 'output[-1]' as the name of the best model, so
    make sure you finish with the token of the model identifiers and nothing else: 'm' or
    'M' (no quotes, no dots, no backticks, no new lines, ...). For example:

### Concise explanation
...some text...

### Which is best, m or M?
M

Now is your turn.

## Your answer: "Concise explanation" followed by "Which is best, m or M?"
```

**CoT Comparison (Tie Allowed)**

```
[System Part]
You are a highly efficient assistant, who evaluates and selects the best large language
    model (LLMs) based on the quality of their responses to a given instruction. This
```

```
    process will be used to create a leaderboard reflecting the most accurate and human-
        preferred answers.
```

**[User Part]**
```
I require a leaderboard for various large language models. I'll provide you with prompts
    given to these models and their corresponding outputs. Your task is to assess these
    responses, and select the model that produces the best output from a human perspective
    . If you determine that both outputs are of equal quality or are unable to decide
    which one is better, you should indicate a tie by providing the identifier 'D'.

## Instruction

{
    "instruction": """{instruction}""",
}

## Model Outputs

Here are the unordered outputs from the models. Each output is associated with a specific
    model, identified by a unique model identifier.

{
    {
        "model_identifier": "m",
        "output": """{output_1}"""
    },
    {
        "model_identifier": "M",
        "output": """{output_2}"""
    }
}

## Task

Evaluate the models based on the quality and relevance of their outputs, and select the
    model that generated the best output. Answer by first providing a concise explanation
    and then end your answer by providing the model identifier of the best output. If you
    determine that both outputs are of equal quality or cannot decide which one is better,
     indicate a tie by using the identifier 'D'. We will use the last character of your
    output 'output[-1]' as the name of the best model, so make sure you finish with the
    token of the model identifiers and nothing else: 'm', 'M'  or 'D' (no quotes, no dots,
     no backticks, no new lines, ...). For example:

### Concise explanation
...some text...

### Which is best, m, M or D?
M

Now is your turn.

## Your answer: "Concise explanation" followed by "Which is best, m, M or D?"
```

**CoT Comparison with Checklist**

**[System Part]**
```
You are a highly efficient assistant, who evaluates and selects the best large language
    model (LLMs) based on the quality of their responses to a given instruction and the
    corresponding criteria. This process will be used to create a leaderboard reflecting
    the most accurate and human-preferred answers.
```

**[User Part]**
```
I require a leaderboard for various large language models. I will provide you with prompts
     given to these models and their corresponding outputs. I will also provide one
    specific evaluation checklist which contains a list of specific criteria that a good
```

```
    output should fulfill. Your task is to assess these responses to see whether they
    satisfy the requirements of the checklist and select the model that produces the best
    output from a human perspective based on the provided checklist.

## Instruction

{
    "instruction": """{instruction}""",
}

## Checklist
Here is the checklist that contains the conditions specified in the question for a good
    output. The more requirements an output meets, the better it is considered.

{
    checklist: """{checklist}""",
}

## Model Outputs

Here are the unordered outputs from the models. Each output is associated with a specific
    model, identified by a unique model identifier.

{
    {
        "model_identifier": "m",
        "output": """{output_1}"""
    },
    {
        "model_identifier": "M",
        "output": """{output_2}"""
    }
}

## Task

Evaluate the models based on the quality and relevance of their outputs, and select the
    model that generated the best output based on the checklist. Answer by first providing
     a concise explanation and then end your answer by providing the model identifier of
    the best output. We will use the last character of your output 'output[-1]' as the
    name of the best model, so make sure you finish with the token of the model
    identifiers and nothing else: 'm' or 'M' (no quotes, no dots, no backticks, no new
    lines, ...). For example:

### Concise explanation
...some text...

### Which is best, m or M?
M

Now is your turn.

## Your answer: "Concise explanation" followed by "Which is best, m or M?"
```

**CoT Comparison with Checklist (Tie Allowed)**

```
[System Part]
You are a highly efficient assistant, who evaluates and selects the best large language
    model (LLMs) based on the quality of their responses to a given instruction and the
    corresponding criteria. This process will be used to create a leaderboard reflecting
    the most accurate and human-preferred answers.

[User Part]
I require a leaderboard for various large language models. I will provide you with prompts
     given to these models and their corresponding outputs. I will also provide one
```

```
      specific evaluation checklist which contains a list of specific criteria that a good
      output should fulfill. Your task is to assess these outputs to see whether they
      satisfy the requirements of the checklist and select the model that produces the best
      output from a human perspective based on the provided checklist. If you determine that
       both outputs are of equal quality or are unable to decide which one is better, you
      should indicate a tie by providing the identifier 'D'.

## Instruction

{
    "instruction": """{instruction}""",
}

## Checklist
Here is the checklist that contains the conditions specified in the question for a good
    output. The more requirements an output meets, the better it is considered.

{
    checklist: """{checklist}""",
}

## Model Outputs

Here are the unordered outputs from the models. Each output is associated with a specific
    model, identified by a unique model identifier.

{
    {
        "model_identifier": "m",
        "output": """{output_1}"""
    },
    {
        "model_identifier": "M",
        "output": """{output_2}"""
    }
}

## Task

Evaluate the models based on the quality and relevance of their outputs, and select the
    model that generated the best output based on the checklist. Answer by first providing
     a concise explanation based on the checklist and then end your answer by providing
    the model identifier of the best output. If you determine that both outputs are of
    equal quality or cannot decide which one is better, indicate a tie by using the
    identifier 'D'. We will use the last character of your output 'output[-1]' as the name
     of the best model, so make sure you finish with the token of the model identifiers
    and nothing else: 'm', 'M'  or 'D' (no quotes, no dots, no backticks, no new lines,
    ...). For example:

### Concise explanation
...some text...

### Which is best, m, M or D?
M

Now is your turn.

## Your answer: "Concise explanation" followed by "Which is best, m, M or D?"
```

### G.2. Checklist Generation

We follow Cook et al. (2024)'s prompt tepmplate to generate checklists.

```
[System Part]
Please help judge an AI assistant's response to an instruction by providing an evaluation
```

```
      checklist.
To write a specific evaluation checklist, you get given the following entity each time:
INSTRUCTION: An instruction that has been given to an AI assistant.
```

**[User Part]**
```
## Task Details
Your task is to come up with an evaluation checklist list for a given INSTRUCTION.
This evaluation checklist should be a list of questions that ask whether or not specific
    criteria relevant to the INSTRUCTION were met by an AI assistant's response.
Criteria covered by your checklist could be explicitly stated in the INSTRUCTION, or be
    generally sensible criteria for the problem domain.
You should, however, try to be concise and not include unnecessary entries in your
    checklist.

Checklist questions should:
- **Be answerable by 'yes' or 'no'**, with 'yes' meaning that the response successfully
    met the corresponding requirement.
- **Be comprehensive, but concise**, meaning that all criteria directly relevant to the
    INSTRUCTION should be represented by a question, but only questions that are very
    clearly relevant should be included.
- **Be precise**, meaning that checklist questions should avoid vague wording and evaluate
     specific aspects of a response, directly using the phrasing of the INSTRUCTION where
    appropriate.

You should always analyse the INSTRUCTION before providing an evaluation checklist.

## Response Format
Analysis: xxx
Answer: CHECKLIST QUESTIONS (each question should appear on a new
line)

## Examples

{examples}

## Real Task

### INSTRUCTION
{message}

### Response
Please analyse the instruction and provide an answer in the correct format.
Remember that each question should be phrased such that answering with 'yes' would mean
    that the response **successfully** fulfilled the criteria being assessed by the
    question.
In most cases, your checklist should contain at least two questions, but no more than
    eight.
```

