# OpenReview forum: "Investigating Non-Transitivity in LLM-as-a-Judge"
_ICML.cc/2025/Conference — ICML 2025 spotlightposter_

### Official Review · Reviewer_rsx8 · 2025-03-04

**Overall Recommendation:** 5

**Summary:**

The authors argue that the existing automated (using LLM-as-a-Judge) LLM ranking algorithms are unreliable and not aligned with human judgement. The authors propose judgement transitivity as a metric of self-consistency to estimate the quality of judgement. In fact, the authors propose two judgement self-consistency metrics: Percentage of Non-Transitive cases (PNT) and the more reliable Soft Non-Transitivity Deviation (SNTD). The authors apply the Bradley-Terry model to reconcile the pairwise rankings into the ranking through all subject LLMs, and convert the Bradley-Terry Coefficients into Elo Rating for further analysis.

The original AlpacaEval resource usage is O(N * M). However, in the first approach the authors have to use the round robin tournament with way higher complexity O(N * M^2).  As all-to-all tournament is very resource consuming, the authors propose Swiss-Wise Iterative Matchmaking (SWIM) tournament with O(N * M * logM ) that implements iterative mining of most unresolved LLM pairs.

**Claims And Evidence:**

The paper closes an important gap in the evaluation of LLMs: understanding and mitigating the deficiencies of LLM-as-a-Judge.

The paper is clearly written, well structured. All claims are supported. The notation, and definitions are clear.

It is clever to use the Bradley-Terry model to infer the overall ranking from pairwise comparisons. To avoid the problem of disjoint ranked groups of models, the authors propose the round-robin tournament setup, and further improve over it with the proposed Swiss-Wise Iterative Matchmaking.

The use of Jensen–Shannon divergence (the symmetrized and smoothed version of KL-divergence) to soften the raw Percentage of Non-Transitive cases (PNT) and arrive to Soft Non-Transitivity is very smart.

The analysis and visualisation of the experimental results is comprehensive. The extensive investigation of the contribution of position bias and judge’s inherent reasoning limitation is noteworthy.

**Essential References Not Discussed:**

There are no essential references that are not discussed.

**Ethical Review Concerns:**

None.

**Experimental Designs Or Analyses:**

Experimental results are sound. Probably the only weak point is that apart from gpt-4-turbo, the weak gpt-3.5 is used as a judge, and it displays high PNT, and, as the authors point out, is weaker than the majority of the LLMs under investigation. It is good to see the results for gpt-3.5-turbo, but its low performance is expected and does not carry much insight.

**Methods And Evaluation Criteria:**

Yes.

**Other Comments Or Suggestions:**

It would be good to see random baselines for PNT and SNTD in Table 1. Not the random choice of the order of the A and B LLM responses in the judge prompt, but the random uniform judge decision.

**Other Strengths And Weaknesses:**

> Violations of transitivity can result in unstable rankings that undermine the evaluation framework’s reliability

It is not clear when non-transitivity can be detrimental in practice, what are the practical situations when it hurts? What is the definition of the evaluation framework reliability?

Lines 132-134
> Given the presence of non-transitivity, evaluating a strategy based on  its performance against a single opponent does not reliably reflect its true capability.

It is not clear what is “true capability”. Who/what is the oracle in this case?

**Questions For Authors:**

None.

**Relation To Broader Scientific Literature:**

The topic of the paper is related to ranking and retrieval.

**Theoretical Claims:**

I checked the expressions conceptually, without detailed verification.

---

> ### Author Rebuttal · Authors · 2025-04-01
>
> We sincerely thank the reviewer for their insightful and positive comments, which significantly enhance the clarity and impact of our manuscript. Please see below for our detailed response.
>
> >  It is good to see the results for gpt-3.5-turbo, but its low performance is expected and does not carry much insight.
>
> We appreciate this valuable feedback from the reviewer, and as such we have conducted additional experiments using GPT-4o-mini (gpt-4o-mini-2024-07-18) as a judge on the AlpacaEval dataset across the same four scenarios setting presented in Table 1. The results are shown below:
>
> ### GPT-4o-mini as the Judge
>
> |Scenario| Models|PNT|SNTD|
> |-|-|-|-|
> | LL| gpt-4o > Qwen1.5-72B > Mistral-7B-Instruct| 3.35 | 0.1006 |
> | LM| gpt-4o > Qwen1.5-72B ≈ Claude-3-Sonnet| 3.60 | 0.1070 |
> | ML| Yi-34B ≈ Qwen1.5-72B > Mistral-7B-Instruct| **3.98** | 0.1036 |
> | MM| Qwen1.5-72B ≈ Claude-3-Sonnet ≈ GPT-4| 3.60 | **0.1173** |
>
> These results support our conclusion that non-transitivity increases as model performance differences narrow, as reflected by SNTD. Furthermore, Chatbot Arena ranks GPT-4o-mini higher than GPT-4-Turbo, suggesting it is a stronger judge. Compared to GPT-4-Turbo, GPT-4o-mini consistently yields lower SNTD and PNT across almost all scenarios which means it’s more transitive, thus further validating the claim that a weaker judge exhibits more non-transitivity.
>
> For comparison, we also provide the earlier GPT-4-Turbo results below:
>
> ### GPT-4-Turbo as the Judge
>
> |Scenario| Models|PNT|SNTD|
> |-|-|-|-|
> | LL| ...| 3.98 | 0.1121 |
> | LM| ... | 5.96 | 0.1336
> | ML| ...| 3.98 | 0.1215 |
> | MM| ...| **8.45** | **0.1431** |
>
> > It is not clear when non-transitivity can be detrimental in practice, what are the practical situations when it hurts? What is the definition of the evaluation framework reliability?
>
> Non-transitivity highlights practical risks in commonly used evaluation frameworks, such as AlpacaEval. Our study empirically demonstrates that selecting different baseline models can yield different model rankings, with only 20% of models maintaining consistent rank positions across various baselines, and pairwise rank agreement drops to 61% on average (Section 4.2). This undermines evaluation reliability, which we define as the framework’s ability to consistently produce stable rankings aligned with human preferences, regardless of baseline choice. Additionally, non-transitivity can lead weaker models to appear superior due to cyclic preferences, potentially causing practitioners to deploy suboptimal models. This risk is particularly critical in applications such as chatbots or automated decision-making systems, where model performance directly impacts user trust and safety.
>
> > It is not clear what is “true capability”. Who/what is the oracle in this case?
>
> We take inspiration from Czarnecki et al. [1] in defining true capability in terms of the vertical (or transitive) component of skill in a distribution of strategies for a given game. They conceptualize the distribution as analogous to a spinning top, where the vertical axis represents skill level (transitive strength), increasing as one moves upward, and the horizontal axis denotes non-transitivity, reflecting strategies’ cyclical relationships (line 127-131). At the widest part of the spinning top, strategies exhibit diverse and strong non-transitive interactions, similar to players with different styles competing against each other. Moving upward, strategies become increasingly homogeneous, and non-transitivity diminishes as skills improve. The "true capability" relates to skill progression toward the Nash Equilibrium, represented by the vertical axis in this analogy. In our work, we approximate the oracle using Elo scores from Chatbot Arena’s crowdsourced rankings, as these are generated through randomized anonymous pairings across diverse user queries, effectively mitigating non-transitivity.
>
> > It would be good to see random baselines for PNT and SNTD in Table 1. Not the random choice of the order of the A and B LLM responses in the judge prompt, but the random uniform judge decision.
>
> We agree with this insightful suggestion and have conducted additional experiments where the judge randomly predicts outcomes with a 50% probability for either preference. Under these conditions, the scenario no longer impacts PNT and SNTD. The results for this random baseline are as follows:
> - PNT: 25
> - SNTD: 0.3465
>
> Note: When calculating SNTD, the default logarithm base is $e$, thus the Jensen–Shannon divergence range is $[0, \log 2] \approx [0, 0.693]$.  Using base 2, the range becomes $[0, 1]$, making the SNTD for a random judge precisely 0.5.
>
> We once again thank the reviewer for their insightful comments, which improve our manuscript significantly.
>
> References:
>
> [1] Czarnecki, W. M., Gidel, G., Tracey, B., Tuyls, K., Omidshafiei, S., Balduzzi, D., & Jaderberg, M. (2020). Real world games look like spinning tops. In NeurIPS.

---

### Official Review · Reviewer_NNcD · 2025-03-10

**Overall Recommendation:** 3

**Summary:**

The paper shows that LLM judges have non-transitive preference in pairwise comparison, which is not only caused by position bias. Furthermore, non-transitivity can be mitigated by round-robin tournaments combined with the Bradley-Terry model. The efficiency can be further improved by Swiss-Wise Iterative Matchmaking.

**Claims And Evidence:**

The claimed contributions 1) and 2) in the paper are supported by experiments on AlpaceEval dataset. The result would be more convincing if more tests are conducted on other datasets, such as Chatbot Arena. This is because the behavior of LLM judges could be different across datasets.

There is no direct experiment or theoretical analysis towards contribution 3), i.e., round-robin tournaments reduces non-transitivity. Correlation is computed in Section 5, but this is not directly related to transitivity of preference.

**Essential References Not Discussed:**

No missing reference found.

**Experimental Designs Or Analyses:**

Strength: The experiment considers multiple factors that could impact non-transitivity of preference.

Weakness: Only AlpacaEval dataset is considered. The behavior of LLM judges could be different on other datasets, e.g., Chatbot Arena.

**Methods And Evaluation Criteria:**

No significant flaws in method and evaluation.

**Other Comments Or Suggestions:**

No additional comments.

**Other Strengths And Weaknesses:**

No additional strength or weakness.

**Questions For Authors:**

1. Why does round-robin tournaments resolves the non-transitivity issue? Is it from a theoretical guarantee or can be demonstrated through experiments?

**Relation To Broader Scientific Literature:**

This paper suggests using round-robin tournaments in LLM-as-a-judge when transitivity is important, contributing to research in LLM-as-a-judge.

**Theoretical Claims:**

Not applicable for this paper.

---

> ### Author Rebuttal · Authors · 2025-04-01
>
> We thank the reviewer for their constructive feedback! To our understanding, the main concerns of the reviewer lie in 1) the limited evaluation on only the AlpacaEval datasets, and 2) the claim that round-robin tournaments reduce non-transitivity – we address both in our response. We hope that the reviewer could considers increasing the score if they feel their concerns have been sufficiently addressed.
>
> > Only AlpacaEval dataset is considered. The behavior of LLM judges could be different on other datasets, e.g., Chatbot Arena.
>
> We appreciate this valuable suggestion from the reviewer and agree that it is crucial to verify the generalizability of our results across different datasets. Consequently, we have conducted additional experiments using GPT-4-Turbo, GPT-3.5-Turbo, and GPT-4o-mini judges on the Arena-Hard-Auto dataset [1], comprising 500 high-quality prompts sourced from Chatbot Arena, with position switching. Models for each scenario are selected based on their rankings in Arena-Hard-Auto's leaderboard. The results are summarized below:
>
> ###  PNT and SNTD for Different Judges in Arena-Hard-Auto
>
> | Scenario | GPT-4-Turbo PNT | GPT-4-Turbo SNTD | GPT-3.5-Turbo PNT | GPT-3.5-Turbo SNTD | GPT-4o-mini PNT | GPT-4o-mini SNTD |
> |----------|------------------|-------------------|--------------------|---------------------|------------------|-------------------|
> | LL       | 2             | 0.0820            | 17              | 0.2071              | 1                | 0.0813            |
> | LM       | 3             | 0.1083            | 17.5              | 0.2002              | 1.5              | 0.0880            |
> | ML       | 2.5             | 0.0945            | 24.5              | **0.2370**              | **5.5**              | 0.1085            |
> | MM       | **5**             | **0.1270**          | **28**             | 0.2294              | 5                | **0.1181**            |
>
> The models evaluated in each scenario are:
> - LL: gpt-4o-2024-05-13 > Qwen1.5-72B-Chat > Mistral-7B-Instruct
> - LM: gpt-4o-2024-05-13 > Mistral-Large-2402 ≈ Qwen1.5-72B-Chat
> - ML: Mistral-Large-2402 ≈ Qwen1.5-72B-Chat > Mistral-7B-Instruct
> - MM: GPT-4-0613 ≈ Mistral-Large-2402 ≈ Qwen1.5-72B-Chat
>
> These supplementary results align with our original findings, reinforcing that non-transitivity generally increases as the performance gap between model pairs narrows with a strong judge, as quantified by the SNTD metric. This consistency suggests that the observed non-transitivity behavior of LLM judges is robust across different datasets. Due to resource constraints, we conducted this analysis on a sample of 200 questions from Arena-Hard-Auto. We hope the reviewer understands that running extensive model evaluations with positional swaps demands substantial computational resources.
>
> > There is no direct experiment or theoretical analysis towards contribution 3), i.e., round-robin tournaments reduces non-transitivity. Correlation is computed in Section 5, but this is not directly related to transitivity of preference […] Why does round-robin tournaments resolves the non-transitivity issue? Is it from a theoretical guarantee or can be demonstrated through experiments?
>
> We respectfully request further clarification on this point, as there may have been a misunderstanding. Our claim is not that round-robin tournaments reduce non-transitivity in judge models—non-transitivity is inherent to the judge and cannot be externally mitigated. Instead, round-robin tournaments reduce the negative impact of non-transitivity. By aggregating pairwise comparisons across all model pairs, this approach avoids reliance on a single baseline model, thereby mitigating the cyclic preferences in the model level caused by judges' inherent non-transitivity. We demonstrate empirically that round-robin tournaments do not suffer from the unreliability demonstrated by baseline-fixed approaches, as round-robin tournaments remove the impact of non-transitivity. While we do not prove this claim theoretically, we believe it would be possible to show that a round-robin tournament measures only the transitive component of skill [2], and hence does not suffer from unreliability due to potentially non-transitive judges.
>
> Once again, we thank the reviewer for their time and detailed feedback. If the reviewer has any further questions or suggestions, we would be more than happy to address them.
>
> References:
>
> [1] Li, T., Chiang, W.-L., Frick, E., Dunlap, L., Wu, T., Zhu, B., Gonzalez, J. E., & Stoica, I. (2024). From crowdsourced data to high-quality benchmarks: Arena-Hard and BenchBuilder pipeline. arXiv preprint arXiv:2406.11939.
>
> [2] Czarnecki, W. M., Gidel, G., Tracey, B., Tuyls, K., Omidshafiei, S., Balduzzi, D., & Jaderberg, M. (2020). Real world games look like spinning tops. In NeurIPS.

---

### Official Review · Reviewer_BZyP · 2025-03-24

**Overall Recommendation:** 4

**Summary:**

This paper explores an issue in comparison-based evaluation: non-transitivity, meaning that in evaluations based on a baseline, if A > B and B > C, it does not necessarily follow that A > C.

The paper first defines how to measure this non-transitivity and establishes a framework for evaluating model performance based on comparisons. The authors then identify the existence of non-transitivity and analyze several influencing factors in detail, including the choice of judge models, the performance gap between compared models, and position bias. These factors make the final ranking results highly sensitive to the choice of the baseline model.

To address this, the paper proposes two improvements. First, it refines the ranking method and introduces a more efficient algorithm. Second, instead of using win rate to represent model performance, it adopts the BT model and ELO rating system for estimation.

**Claims And Evidence:**

Yes, the authors experimentally prove the existence of non-transitivity and analyze the influence factor of non-transitivity. They also demonstrate the effectiveness of the new methods through experiments.

**Essential References Not Discussed:**

To the best of my knowledge, the paper adequately covers all the essential related works necessary for understanding its key contributions.

**Experimental Designs Or Analyses:**

Mostly good. However, since the authors only select GPT-4 and GPT-3.5 as judges, the robustness of the conclusion "Weaker Judge is More Non-Transitive" is limited. This has been pointed out in their limitations.

**Methods And Evaluation Criteria:**

Yes, the selection of AlpacaEval and LLMs is appropriate. The Spearman and Kendall correlation are suitable to compare the AlpacaEval and Chatbot Area.

**Other Comments Or Suggestions:**

line 167: currying -> carrying
line 172: (M-1) -> M. when a new model are added, it need to compare with existing M models in round-robin tournament.

It is recommended to carefully review and correct any remaining unnoticed minor errors.

**Other Strengths And Weaknesses:**

Strengths: This paper explores non-transitivity in a systematic manner, starting from its formal definition, followed by result analysis, and finally proposing improvements.

Weaknesses: There are a few minor errors, including the derivation errors mentioned above and several spelling mistakes mentioned below.

**Questions For Authors:**

None.

**Relation To Broader Scientific Literature:**

The paper's key contribution is showing that non-transitivity, previously observed in other zero-sum games, also applies to LLMs.

**Theoretical Claims:**

The derivation of equation 5 is wrong, the correct derivation is as follows: $$\phi(o_A^{(i)}, o_B^{(i)} \mid m_J, I_i) = \frac{1}{1 + e^{-(\gamma_A^{(i)} - \gamma_B^{(i)})}} = \frac{1}{1 + e^{-((\gamma_A^{(i)} - \gamma_C^{(i)}  )- (\gamma_B^{(i)}- \gamma_C^{(i)} ))}} = \frac{1}{1 + e^{-(e_{AC}^{(i)} - e_{BC}^{(i)})}} =  \frac{1}{1 + e^{e_{BC}^{(i)} - e_{AC}^{(i)}}} $$

---

> ### Author Rebuttal · Authors · 2025-04-01
>
> We sincerely appreciate the reviewer’s thoughtful and constructive feedback, which greatly helps us strengthen our manuscript. Below, we provide detailed responses addressing each of the reviewer’s comments:
>
> > …since the authors only select GPT-4 and GPT-3.5 as judges, the robustness of the conclusion "Weaker Judge is More Non-Transitive" is limited. This has been pointed out in their limitations.
>
> We appreciate the reviewer highlighting the importance of robustness in verifying the conclusion "Weaker Judge is More Non-Transitive.", and as such we have conducted additional experiments using GPT-4o-mini (gpt-4o-mini-2024-07-18) as a judge on the AlpacaEval dataset across the same four scenarios setting presented in Table 1. The results are shown below:
>
> ### GPT-4o-mini as the Judge
>
> |Scenario| Models|PNT|SNTD|
> |-|-|-|-|
> | LL       | gpt-4o > Qwen1.5-72B > Mistral-7B-Instruct                              | 3.35 | 0.1006 |
> | LM       | gpt-4o > Qwen1.5-72B ≈ Claude-3-Sonnet                                  | 3.60 | 0.1070 |
> | ML       | Yi-34B ≈ Qwen1.5-72B > Mistral-7B-Instruct                              | **3.98** | 0.1036 |
> | MM       | Qwen1.5-72B ≈ Claude-3-Sonnet  ≈ GPT-4                                   | 3.60 | **0.1173** |
>
> These additional results remain consistent with our previous conclusion, indicating that the degree of non-transitivity generally increases as the performance gap between model pairs narrows, which is confirmed by the SNTD metric. Furthermore, according to rankings from Chatbot Arena [1], GPT-4o-mini is ranked higher than GPT-4-Turbo, suggesting that GPT-4o-mini serves as a stronger judge. Compared to GPT-4-Turbo, GPT-4o-mini consistently yields lower SNTD and PNT across almost all scenarios which means it is more transitive, thus further validating the claim that a weaker judge exhibits more non-transitivity.
>
> For comparison, we also provide the earlier GPT-4-Turbo results below:
>
> ### GPT-4-Turbo as the Judge
>
> |Scenario| Models|PNT|SNTD|
> |-|-|-|-|
> | LL       | gpt-4o > Qwen1.5-72B > Mistral-7B-Instruct                              | 3.98 | 0.1121 |
> | LM       | gpt-4o > Qwen1.5-72B ≈ Claude-3-Sonnet                                  | 5.96 | 0.1336 |
> | ML       | Yi-34B ≈ Qwen1.5-72B > Mistral-7B-Instruct                              | 3.98 | 0.1215 |
> | MM       | Qwen1.5-72B ≈ Claude-3-Sonnet ≈ GPT-4                                   | **8.45** | **0.1431** |
>
>
> > The derivation of equation 5 is wrong
>
> We sincerely thank the reviewer for pointing out this mistake. We confirm that this is indeed a typographical error, where we miss a minus sign before $(s_{AC}^{(i)} - s_{BC}^{(i)})$. However, our actual implementation uses the correct formulation, as verifiable by the `estimate_win_rate` function within the `check_bias.ipynb` file provided in our Supplementary Material. Specifically, defining:
> $$ X = \phi(o_A^{(i)}, o_B^{(i)} \mid m_J, I_i), \quad Y = \phi(o_B^{(i)}, o_C^{(i)} \mid m_J, I_i), \quad Z = \phi(o_A^{(i)}, o_C^{(i)} \mid m_J, I_i). $$
> According to the Bradley-Terry model, we have:
> $$
> s_{AC}^{(i)} = \gamma_A^{(i)} - \gamma_C^{(i)} = \ln\left( \frac{Z}{1 - Z} \right), \quad
> s_{BC}^{(i)} = \gamma_B^{(i)} - \gamma_C^{(i)} = \ln\left( \frac{Y}{1 - Y} \right), \quad
> s_{AC}^{(i)} - s_{BC}^{(i)} = \ln\left( \frac{Z(1 - Y)}{Y(1 - Z)} \right).
> $$
> When substituted back into  $\hat{\phi}(o_A^{(i)}, o_B^{(i)} \mid m_J, I_i) = \frac{1}{1 + e^{-(s_{AC}^{(i)} - s_{BC}^{(i)})}},$
> the equation becomes $\frac{1}{1 + \exp\left[-\ln\left(\frac{Z(1 - Y)}{Y(1 - Z)}\right)\right]} = \frac{Z(1 - Y)}{(Y + Z) - 2YZ},$
> which aligns precisely with our implementation in `estimate_win_rate_X`. Thus, this typographical error does not affect our experimental conclusions. We have corrected this in the updated manuscript and would like to again thank the reviewer for their diligence.
>
> > line 167: currying -> carrying
>
> "Currying" here refers to a concept in functional programming, denoting the process of fixing one argument of a function to create a new function with fewer arguments.
>
> > line 172: (M-1) -> M. when a new model are added, it need to compare with existing M models in round-robin tournament.
>
> We thank the reviewer for identifying this typographical mistake. Indeed, it should be $M$ comparisons rather than $(M-1)$. This has been corrected in our updated manuscript.
>
> We once again thank the reviewer for their valuable comments, which significantly improve the manuscript's clarity and rigor.
>
> References:
>
> [1] Wei-Lin Chiang, Lianmin Zheng, Ying Sheng, Anastasios N. Angelopoulos, Tianle Li, Dacheng Li, Banghua Zhu, Hao Zhang, Michael I. Jordan, Joseph E. Gonzalez, and Ion Stoica. 2024. Chatbot arena: an open platform for evaluating LLMs by human preference. In Proceedings of the 41st International Conference on Machine Learning (ICML'24), Vol. 235. JMLR.org, Article 331, 8359–8388.

---

> > ### Comment · Reviewer_BZyP · 2025-04-04
> >
> > Thank you for the detailed response and I will maintain my positive score.

---

### Official Review · Reviewer_bBXd · 2025-03-25

**Overall Recommendation:** 4

**Summary:**

This paper investigates the assumption of transitive preferences in LLM-based evaluation frameworks. The authors highlight that non-transitivity exists in LLM judgments, leading to inconsistencies in model rankings depending on the choice of baseline. The authors propose using round-robin tournaments combined with the Bradley-Terry model to generate more robust rankings. Additionally, they introduce Swiss-Wise iterative Matchmaking (SWIM) tournaments to reduce computational costs while preserving ranking reliability. The proposed methods improve correlation with human evaluation benchmarks such as Chatbot Arena.

**Claims And Evidence:**

- Extensive empirical evaluation using AlpacaEval supports the existence of non-transitivity in LLM-as-a-Judge frameworks, demonstrating inconsistencies in model rankings when changing baseline models.
- The study focuses on AlpacaEval as the primary benchmark. Other testing on additional datasets like MT-Bench or WildBench would strengthen generalizability.

**Essential References Not Discussed:**

The authors paid special attention to the existence of non-transitivity in the real world.

As a former literature, the Bradley-Terry(BT) model has been known to be exposed to 'intransitivity' risk because it relies on scalar variables, where all preferences are transitive by assumption.
- The literature below studied representative preference datasets in the real world, where the 'transitive' relationship between preference annotations may not always hold.
- https://arxiv.org/abs/2409.19325 (Duan et al, 2017)
- Besides evidence and quantitative evaluation of non-transitivity, the paper proposed representation learning algorithms to generalize BT models to a 'non-transitive' setting. To my knowledge, this can be considered related work, and representation learning techniques (profiling of LLMs) are still under-explored in the LLM-as-a-judge topic.

**Experimental Designs Or Analyses:**

- The experiments are generally sound, with a high level of attention to detail.
- The authors used position switching to control and avoid ordering bias.
- To mitigate verbosity bias and ensure a fair comparison, the authors adopt the generalized linear model with the same weights as Length-Controlled AlpacaEval.
- The study does not include explicit human verification of non-transitive cases. While Chatbot Arena provides a reference ranking, a small-scale micro study verification of LLM judgments would add credibility.

**Methods And Evaluation Criteria:**

- The authors defined the Percentage of Non-Transitive cases (PNT) and Soft Non-Transitivity Deviation (SNTD) to measure the degree of non-transitivity for a single instruction with a triplet of models. This is a meaningful effort to quantify the existence of non-transitivity.
- The arguments and remarks give practical guidance for the problem by comparing it to human preference rankings from Chatbot Arena.

**Other Comments Or Suggestions:**

See above.

**Other Strengths And Weaknesses:**

- This paper highlights the existence of 'non-transitivity' in LLM-as-a-judge applications.
- The proposals in the work, both evaluation metrics and algorithms are simple and relative to existing approaches.
- This paper effectively connects ranking instability to non-transitive behavior.

**Questions For Authors:**

See above.

**Relation To Broader Scientific Literature:**

RM is the foundational pillar in RLHF and LLM-as-a-judge emerges as the standard tool for LLM evaluation in broader scientific applications.

**Theoretical Claims:**

This paper does not have formal theoretical claims but rather focuses on the foundation problem in prevailing RMs.

---

> ### Author Rebuttal · Authors · 2025-04-01
>
> We sincerely thank the reviewer for their positive and insightful comments. These suggestions have significantly enhanced the clarity and depth of our manuscript. Below, we address each concern raised in detail.
>
> > The study does not include explicit human verification of non-transitive cases. While Chatbot Arena provides a reference ranking, a small-scale micro study verification of LLM judgments would add credibility.
>
> We appreciate the reviewer for highlighting the importance of explicit human verification of non-transitive cases. Although human verification is valuable, it is beyond the immediate scope of this study, as our central objective is to investigate non-transitivity specifically arising from LLM-based judgments. In other words, the phenomenon of interest is the inherent non-transitivity within LLM evaluators themselves which is independent of human judgment. While it would indeed be intriguing to examine whether similar patterns also occur in human evaluators, such an investigation represents an exploration of related but distinct phenomena rather than a direct validation of LLM judgments. Nevertheless, we recognize the significance of this point and have noted this as an important direction for future work in the updated manuscript, indicating our intention to perform a targeted micro-study to explore alignment or divergence between LLM and human judgments in non-transitive scenarios.
>
> > As a former literature, the Bradley-Terry(BT) model has been known to be exposed to 'intransitivity' risk because it relies on scalar variables, where all preferences are transitive by assumption.
> > - The literature below studied representative preference datasets in the real world, where the 'transitive' relationship between preference annotations may not always hold.
> > - https://arxiv.org/abs/2409.19325 (Duan et al, 2017)
> > - Besides evidence and quantitative evaluation of non-transitivity, the paper proposed representation learning algorithms to generalize BT models to a 'non-transitive' setting. To my knowledge, this can be considered related work, and representation learning techniques (profiling of LLMs) are still under-explored in the LLM-as-a-judge topic.
>
> We appreciate the reviewer’s insightful comment regarding the inherent assumption of transitivity in both the Elo and Bradley-Terry models. We choose the BT model primarily because, despite known cyclic behaviors observed in practice (e.g., in competitive games such as StarCraft II, and Dota 2), their transitivity property is still considered valid for comparative ranking purposes [1] and that is why ELO scores have remained widely used to evaluating agents in non-transitive games [2, 3]. While our study indeed observes non-transitivity at the instance level when using GPT-4-Turbo as the judge, these instances are relatively infrequent, meaning that aggregated model-level evaluations remain predominantly transitive. Consequently, the observed non-transitivity introduces only mild disturbances, effectively manageable within the BT framework. Nevertheless, we fully acknowledge the reviewer’s point that the transitivity assumption in the BT model may not fully capture the nuanced capabilities of models. We have revised the related work section to include this discussion, highlighting representation learning techniques, such as those presented by Duan et al. (2017), as promising and still under-explored methods that could enhance the robustness of LLM-as-a-judge evaluations.
>
> Once again, we thank the reviewer for their time and detailed feedback. We hope these clarifications address the reviewer’s concerns and welcome any additional feedback to further improve our work.
>
> References:
>
> [1] Bertrand, Q., Czarnecki, W. M., & Gidel, G. (2023). On the limitations of the Elo, real-world games are transitive, not additive. Proceedings of The 26th International Conference on Artificial Intelligence and Statistics (AISTATS), PMLR 206: 2905–2921.
>
> [2] Vinyals, O., Babuschkin, I., Czarnecki, W.M. et al. Grandmaster level in StarCraft II using multi-agent reinforcement learning. Nature 575, 350–354 (2019).
>
> [3] Siqi Liu et al. ,From motor control to team play in simulated humanoid football.Sci. Robot.7,eabo0235(2022).

---

> > ### Comment · Reviewer_bBXd · 2025-04-08
> >
> > Thank you for the detailed clarification regarding the concerns raised. I will maintain my positive score.

---

### Official Review · Reviewer_Dskp · 2025-04-03

**Overall Recommendation:** 4

**Summary:**

The paper investigates whether LLM exhibit non-transitive preferences when comparing model outputs. Typically, people use pairwise comparisons against a single baseline model, implicitly assuming transitive preferences. However, the authors find that such judgments can violate transitivity and that rankings can change significantly if a different baseline is used. They propose measuring soft non-transitivity deviation, mitigating position bias by switching model-response order, and moving to a round-robin tournament framework combined with Bradley-Terry scoring. This approach reduces the influence of non-transitivity and aligns more closely with human-preference rankings while being more robust than baseline-fixed methods. They further introduce a computationally lighter Swiss-Wise Iterative Matchmaking method for a cheaper round-robin.

**Claims And Evidence:**

The claim that Non-transitive preferences exist in LLM-based evaluations is largely supported. Position bias partially drives non-transitivity is also supported (this fact is not very surprising). Baseline-fixed methods, such as only compare to GPT-4, yield rankings sensitive to which baseline is chosen is also supported (this fact also not very surprising). Round-robin tournaments plus Bradley-Terry modeling improve reliability and correlate more strongly with human judgments is supported.

**Essential References Not Discussed:**

I don't think there aren't any essential references not discussed.

**Experimental Designs Or Analyses:**

The experiments are largely well designed to support most of the claims.

One limitation is that the data come mostly from AlpacaEval, which may not capture all open-ended domains. Raises question regarding generalizability.

Another limitation is that System Prompt is a big factor in LLM-as-a-Judge framework that isn't studied in the work. It would be nice if the paper further explore the effects of System Prompt to transitivity (e.g., AlpacaEval system prompt vs MT Bench system prompt vs Arena-Hard system prompt).

**Methods And Evaluation Criteria:**

Evaluation primarily relies on comparing final rankings to Chatbot Arena’s human-preference ordering, a recognized baseline for alignment with human judgments.

**Other Comments Or Suggestions:**

N/A

**Other Strengths And Weaknesses:**

Strength:
- Thorough empirical analysis of a largely overlooked concern—non-transitive comparisons. I like the author's work, I think more papers like this will improve the LLM-as-a-Judge framework.
- Practical strategies introduced.
- Good presentation.

Weakness:
- As I mentioned above in "Experimental Designs Or Analyses".

**Questions For Authors:**

Would you guys release the code / implementation for the SWIM method? I think it will benefit the automatic evaluation community.

**Relation To Broader Scientific Literature:**

It relates to recent work on LLM biases (position bias, verbosity bias) and automated evaluators, highlighting that pairwise comparisons can inherit these biases and yield inconsistent rankings.

**Theoretical Claims:**

I did not find any incorrectness.

---

### Decision · Program_Chairs · 2025-05-01

**Decision:**

Accept (spotlight poster)

**Comment:**

The paper investigates non-transitivity in LLM-as-a-Judge frameworks, proposing round-robin tournaments and SWIM to address ranking inconsistencies. Reviews highlighted several strengths in
1. Novel Contribution: Identifies and quantifies non-transitivity in LLM evaluations, a critical issue for reliable benchmarking.
2. Empirical Rigor: Comprehensive experiments on AlpacaEval and Arena-Hard-Auto, with position bias controls and multiple judges (GPT-4-Turbo, GPT-3.5, GPT-4o-mini), some of those from additional experiments in the rebuttal messages.

The paper makes a significant contribution to LLM evaluation by exposing and addressing a fundamental issues of LLM-as-a-Judge nontransitivity.  The proposed PNT/SNTD and SWIM offer practical tools for the community.

The authors also mostly adequately address the reviewers' concern, further strengthening the paper's message.